# Model of neural induction in the ascidian embryo

**Rossana Bettoni**[1,2,3], **Clare Hudson**[4]*, **Géraldine Williaume**[4], **Cathy Sirour**[4], **Hitoyoshi Yasuo**[4], **Sophie de Buyl**[2,3‡], **Geneviève Dupont**[1,3‡]*

**1** Unité de Chronobiologie Théorique, Faculté des Sciences, CP231, Université Libre de Bruxelles (ULB), Boulevard du Triomphe, Brussels, Belgium, **2** Applied Physics Research Group, Vrije Universiteit Brussel, Brussels, Belgium, **3** Interuniversity Institute of Bioinformatics in Brussels, ULB-VUB, La Plaine Campus, Brussels, Belgium, **4** Laboratoire de Biologie du Développement de Villefranche-sur-Mer, Institut de la Mer de Villefranche-sur-Mer, Sorbonne Université, CNRS, Villefranche-sur-Mer, France

‡ Co-last authors.
* clare.hudson@imev-mer.fr (CH); Genevieve.Dupont@ulb.be (GD)

**Data Availability Statement:** Codes are available at https://github.com/rossanabettoni/Model-of-neural-induction-in-the-ascidian-embryo.

**Funding:** RB is supported by a FRIA fellowship. GD is Research Director at the Belgian "Fonds National

## Abstract

How cell specification can be controlled in a reproducible manner is a fundamental question in developmental biology. In ascidians, a group of invertebrate chordates, geometry plays a key role in achieving this control. Here, we use mathematical modeling to demonstrate that geometry dictates the neural-epidermal cell fate choice in the 32-cell stage ascidian embryo by a two-step process involving first the modulation of ERK signaling and second, the expression of the neural marker gene, *Otx*. The model describes signal transduction by the ERK pathway that is stimulated by FGF and attenuated by ephrin, and ERK-mediated control of *Otx* gene expression, which involves both an activator and a repressor of ETS-family transcription factors. Considering the measured area of cell surface contacts with FGF- or ephrin-expressing cells as inputs, the solutions of the model reproduce the experimental observations about ERK activation and *Otx* expression in the different cells under normal and perturbed conditions. Sensitivity analyses and computations of Hill coefficients allow us to quantify the robustness of the specification mechanism controlled by cell surface area and to identify the respective role played by each signaling input. Simulations also predict in which conditions the dual control of gene expression by an activator and a repressor that are both under the control of ERK can induce a robust ON/OFF control of neural fate induction.

## Author summary

The development of a single cell zygote into a multicellular embryo occurs thanks to the combination of cell division and cell specification. The latter process corresponds to the progressive acquisition by the embryonic cells of their final physiological and functional characteristics, which rely on well-defined signaling-controlled genetic programs. The origin of the great robustness of cell specification remains poorly understood. Here, we address this question in the framework of the embryonic neural fate induction in

pour la Recherche Scientifique" (FRS-FNRS) and acknowledges financial support from the ARC project "Noise sensitivity of gene regulatory networks underlying cell fate specification" financed by the Université libre de Bruxelles (ULB). The team of HY is supported by the Centre National de la Recherche Scientifique (CNRS), Sorbonne University, the Fondation ARC pour la Recherche sur le Cancer (PJA 20131200223) and the Agence Nationale de la Recherche (ANR-17-CE13-0003-01). The team of HY acknowledges support from the imaging platform (PIM, member of MICA) and animal facility (CRB) that are funded by the EMBRC-France, whose French funds are managed by the ANR within the Investments of the Future program under reference ANR-10-INBS-0. The funders had no role in study design, data collection and analysis, decision to publish, or preparation of the manuscript.

**Competing interests:** The authors have declared that no competing interests exist.

ascidians, which are marine invertebrates. At the 32-cells stage, four cells identified by their precise location in the embryo adopt neural fate. On the basis of experimental observations, we develop a mathematical model that predicts that the choice between the neural or epidermal fate is controlled by the cell surface areas of the cells in contact with two antagonistic signals, FGF and ephrin. Our findings provide a computational confirmation of the major role played by the geometry of the embryo in controlling cell lineage acquisition during ascidian development.

## Introduction

Embryonic development is a reproducible process during which cells adopt different identities with a high spatiotemporal precision. This precision relies on the ability of cells to interpret signals from their environment and activate specific genetic programs. An accurate description of the interplay between signaling and transcriptional outputs is thus required to gain a clear mechanistic knowledge of cell fate determination in embryonic development. In this context, mathematical models provide useful tools to develop a unifying framework to describe various types of experiments performed to investigate the relation between signaling and gene expression. Models can also answer questions that cannot be readily investigated experimentally [1,2].

Ascidians are marine invertebrate chordates belonging to the subphylum Tunicata, a sister group of vertebrates. Although the ascidian tadpole larvae exhibit the basic chordate body plan, these embryos are much simpler than vertebrates. Moreover, the embryogenesis of ascidians proceeds with an invariant cell division pattern, such that cellular configurations and cell cycle progression are quasi-invariant [3,4]. This allows precise identification of cells with known developmental outcomes. By combining experimental observations and computational modeling, Guignard et al. [4] predicted that the cell contact areas between signal emitting and signal receiving cells play a primary role in determining cell fates during cleavage and gastrula stage ascidian embryos. Geometric control of embryonic inductions contrasts with the role of morphogen gradients described in other systems, in which ligand concentrations determine cellular responses [1,5].

In the present study, we focus on the neural-epidermal binary fate choice that takes place within the ectoderm field of the 32-cell stage ascidian embryo. This onset of neural induction is recognized by expression of the *Otx* gene. *Otx* is activated as an immediate-early response to the extracellular-signal-regulated kinase (ERK) pathway, downstream of fibroblast growth factor (FGF). ERK directly activates *Otx* expression via the Ets1/2 transcription factor [6]. While all 16 ectoderm cells are in direct contact with FGF-expressing mesendoderm cells (Fig 1B) and are competent to respond to FGF, only four specific cells exhibit ERK activation levels sufficient to induce expression of *Otx* and thus adopt neural fate. The precision of this response depends on ephrin/Eph signaling taking place between ectoderm cells themselves [7]. We have recently shown that each ectoderm cell exhibits a level of ERK activation that correlates with its area of surface contact with FGF-expressing mesendoderm cells, while ephrin/Eph signals are important to dampen ERK activation across all ectoderm cells [8]. In contrast to the situation encountered in *Xenopus laevis* oocyte maturation for example [9], during ascidian neural induction, ERK activation is not an all-or-none process but depends on the area of cell surface contact with FGF-expressing cells in a slightly non-linear manner. The graded ERK activation response is converted into a bimodal transcription output of *Otx*, which is restricted to neural precursors. The transcription factor ERF2, also under the control of ERK activity, represses

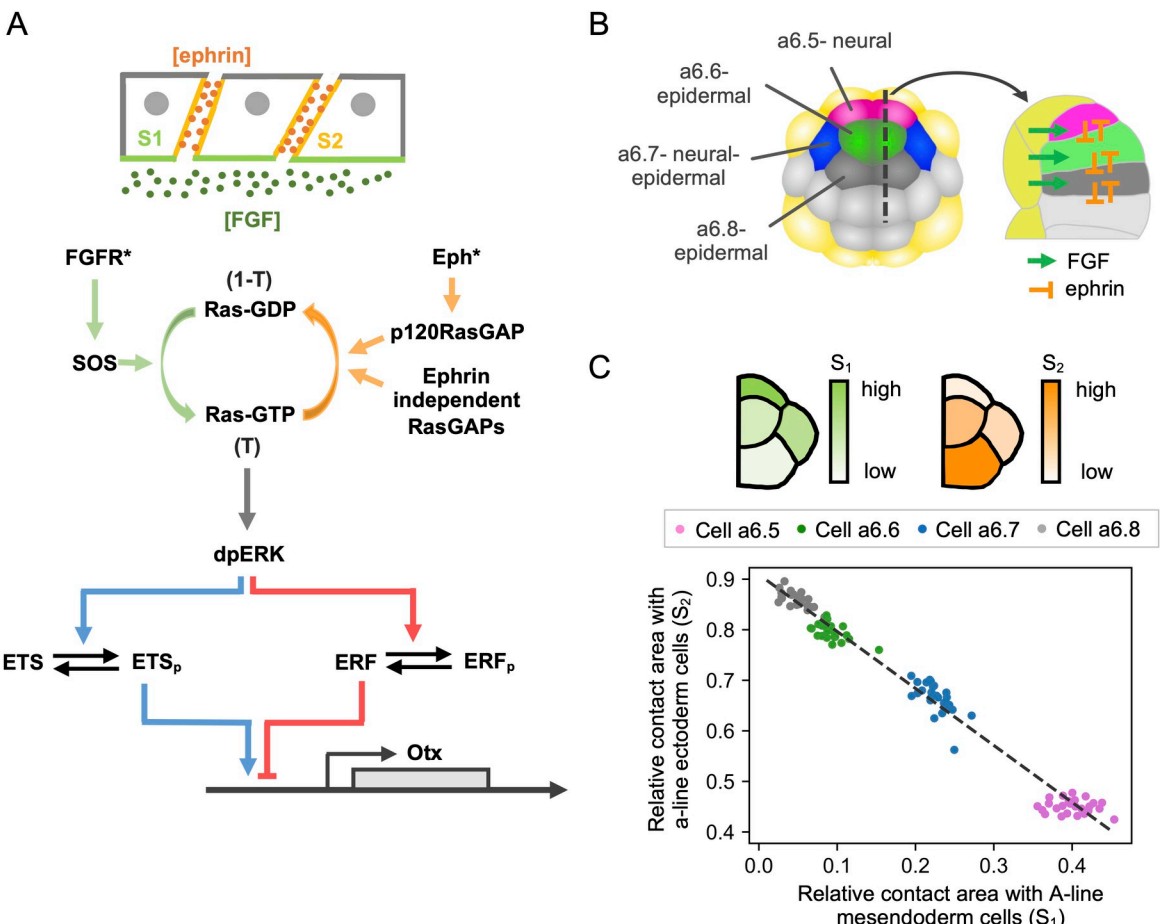

**Fig 1. Model presentation. (A)** a-line ectoderm cells of the ascidian embryo have different cell contact surfaces with FGF-expressing mesendoderm cells and with ephrin-expressing ectoderm cells. FGF binding activates the FGF receptor. Activated FGF receptors (FGFR*) then recruits the guanine nucleotide exchange factor SOS, which promotes the conversion of Ras-GDP to Ras-GTP. At the same time, the binding of the ephrin ligand, Efna.d, activates the Eph receptor. The active Eph receptor (Eph*) in turn recruits p120RasGAP, which promotes the conversion of Ras-GTP to Ras-GDP. Ras-GTP is also converted to Ras-GDP by ephrin-independent RasGAPs. Ras-GTP induces the activation (double phosphorylation: dp) of the extracellular signal-regulated kinase ERK via a kinase cascade including Raf and MEK. Phosphorylation of the ETS1/2 transcription factor by dpERK enhances the expression of the neural marker gene *Otx*. At the same time, dpERK phosphorylates the Ets repressor factor 2 (ERF2, indicated as ERF in the figure), which inhibits *Otx* expression. Figure adapted from [8]. **(B)** Left: Scheme of the 32-cell stage ascidian embryo. a-line ectoderm cells are shown in different colors: a6.5 cell in magenta, a6.6 cell in green, a6.7 cell in blue, and a6.8 cell in gray. Mesendoderm cells are shown in yellow and b-line ectoderm cells in light grey. The dotted line is the line of sagittal section used for the right panel. Right: sagittal section highlighting cell surface contacts between ectoderm and mesendoderm cells and signals considered in the model. Figure adapted from [8]. **(C)** Top: Scheme of the a-line ectoderm cells, showing in different colors the area of each cell surface in contact with FGF expressing cells ($S_1$, on the left) and with ephrin expressing cells ($S_2$, on the right). Bottom: Relative area of a-line ectoderm cell surface in contact with A-line mesendoderm cells, $S_1$, and ectoderm cells, $S_2$, for the four cell types indicated in panel B. The relative contact area is computed as the surface contact /total cell surface. Each dot represents a single cell. Fitting the experimental data with a linear function (shown as a black dashed line), we obtained the expression for the relation between $S_1$ and $S_2$: $S_2 = -1.13 S_1 + 0.91$ ($R^2 = 0.9896$). Figure redrawn from [8].

*Otx* expression in low-ERK cells. In consequence, the dual control exerted by ERK activity on the Ets1/2-mediated activation and ERF2-mediated repression of *Otx* expression appears to play a central role in generating ultrasensitivity in the ERK-*Otx* relationship, which allows neural induction to be an ON or OFF process [8].

In our previous work [8], we used mathematical modeling to enhance the interpretation of the data and draw conclusions about some specific parts of the molecular mechanism underlying neural induction in the 32-cell stage ascidian embryo. In particular, minimal and non-

parametrized modeling allowed the investigation of the role of cell surface contact *versus* ligand concentration in the ERK activation status, as well as the respective roles of the antagonistic enzymes SOS and p120RasGAP that control the regulatory status of Ras, the entry point for the ERK signaling cascade. A theoretical analysis of the dual control of *Otx* expression by an activator and a repressor, both under the control of ERK activity, was also performed to investigate the possible effect of a repressor on the ERK-*Otx* relation. Here, we combine the different mathematical modules developed previously to provide an enhanced, full mathematical description of FGF and ephrin-signaling control of *Otx* expression during early neural induction in the 32-cell stage ascidian embryo. The model, which consists of a set of ordinary differential equations, is solved at steady state. Its solutions are compared with experimental observations mainly derived from our previous work [8]. Results indicate that, using a single set of parameters, the model can adequately account for experimental observations related to neural induction when cells of the ascidian embryo are supposed to differ only by their cell surface contacts with FGF- and ephrin-expressing cells. The same set of parameter values can account for observations performed on pharmacologically or experimentally perturbed embryos. The robustness of the model is corroborated by sensitivity analysis. Finally, the model is used to make theoretical predictions by exploring relations that have not yet been obtained experimentally, for example the dependence of *Otx* expression on the cell surface area in contact with the signaling molecules in the absence of the ERF2 repressor.

## Results

### Overview of the model

The model considers that each cell perceives a level of FGF signaling that is proportional to the area of cell contact with FGF-expressing mesendoderm cells and a level of ephrin signaling that is proportional to the area of cell contact with ephrin-expressing ectoderm cells, as schematized in Fig 1A. These surfaces have been quantified (Fig 1C) and are used to determine the number of receptors in contact with the signals. Ligand-bound FGF receptors (FGFR) and ephrin-bound Eph receptors stimulate the activity of SOS and p120RasGAP, respectively. SOS transforms Ras-GDP into Ras-GTP. Ras-GTP is converted back into Ras-GDP by p120Ras-GAP and by a cell surface contact-independent GAP activity. Ras-GTP activates the ERK signaling cascade [10,11]. The relation between ERK activity and the level of Ras-GTP is described by a single function, which represents the full Ras-Raf-MEK-ERK pathway. Active ERK in turn phosphorylates the two transcription factors, Ets1/2 and ERF2. Phosphorylated Ets1/2 promotes the expression of the neural immediate-early gene *Otx*, whereas unphosphorylated ERF2 represses ERK targets [6,12,13]. The model consists in 5 ordinary differential and 13 algebraic equations describing the abovementioned phenomena using Michaelis-Menten and Hill type kinetic expressions. A full description of the model is provided in the Modeling section.

In the 32-cell stage ascidian embryo, neural induction takes place in the ectoderm lineages (anterior (a) and posterior (b) lines) and is dependent on cell-cell contact with FGF-expressing mesendoderm cells (anterior (A) and posterior (B) lines). We mainly focus on the four pairs of a-line cells, namely a6.5, a6.6, a6.7, and a6.8 in the anterior ectoderm (see Fig 1B; [8]). These four cells display different relative areas of cell surface contact with A-line mesendoderm cells that express FGF, with the neural precursor, a6.5, having the largest area of cell surface contact (Fig 1C). Areas of cell surface contact with FGF-expressing cells and ephrin-expressing cells are inversely correlated, such that cells having the largest cell surface contact with FGF-expressing cells have the lowest cell surface contact with ephrin-expressing ectoderm cells (Fig 1C). In the next section, we model ERK activation and *Otx* expression in these four cells.

## Regulation of ERK activity by fractions of cell surfaces exposed to antagonistic signals

Experimental evidence indicates that, in the 32-cell stage ascidian embryo, the level of ERK activation in ectoderm cells is controlled by the extent of cell surface in contact with FGF-expressing mesendoderm cells and with ephrin-expressing ectoderm cells [8]. Because these two contact areas are correlated, the area of cell surface contact with FGF-expressing cells (designated $S_1$) can also be used to define the area of cell surface contact with ephrin-expressing cells (designated $S_2$) (Fig 1C). Thus, in terms of modeling, the only parameter that is cell-type-specific is $S_1$. The steady states solutions of Eqs (1–13) should correspond to the levels of ERK activation measured in the different cell types as anti-dpERK immunofluorescence (IF) signals in their nucleus. Importantly, extracellular FGF and ephrin concentrations are assumed to be the same for all cell types. As shown in Fig 2A, good agreement between observed and modeled ERK activation levels can be obtained for the four cell types when considering the values of parameters listed in Table 1 and assuming a linear relationship between computed normalized ERK activity (Erk*) and dpERK IF signal measured experimentally (Eq (14)).

The computed Erk* activities are equal to 0.0021 ± 0.0013, 0.0078 ± 0.0038, 0.0551 ± 0.0096 and 0.1787 ± 0.0201 in the a6.8, a6.6, a6.7 and a6.5 cell types, respectively (S1A Fig). Each value was computed as the average between the steady state solutions obtained for the 25 values of the area of surface contact ($S_1$) measured experimentally. Computed values reproduce the observed moderately nonlinear relation between ERK activation and cell surface in contact with FGF-expressing mesendoderm cells. The nonlinear relationship between $S_1$ and ERK activation can also be compared with experiments where cell surfaces and dpERK were both measured at the individual cell level (Fig 2B, $n_H$ = 2.41). The model predicts a modest level of ERK activation, defined between 0 and 1, in the maximally activated a6.5 cell, ~0.18 (S1A Fig). This value stems from the assumed relatively low concentration of extracellular FGF ($K_D/5$) and, to a lesser extent, from the presence of ephrin (see below).

The model was next challenged to predict the level of ERK activation in b-line cells. b-line cells are in contact both with A-line mesendoderm cells that express a uniform level of the ligand *FGF9/16/20* and with B-line mesendoderm cells that express different levels of *FGF9/16/20* (S2A Fig) [8]. In order to account for different levels of FGF expression, equations consider the existence of two distinct populations of FGF receptors, as described in the Modeling section (see section '*Model of Erk activity in b-line cells*'). Model predictions recapitulate experimental observations (S2B Fig) keeping the same set of values of parameters. Only b6.5 cell displays a high level of ERK activity, which is mainly due to the large population of FGF receptors in contact with A-line mesendoderm cells.

In order to fit the computed values to the experimental ones, the fluorescence background needs to be adjusted (see Modeling section). Much of the uncertainty when comparing Erk* to measured dpERK IF signals stems from the background level of fluorescence in experimental measurements. Experimentally determined values are scattered around computed ones (Fig 2). It remains unknown how much of the variation in measured ERK levels within each cell type are true to the system or are inherent to the experimental technique. The larger dispersion of experimental than theoretical values may reveal the existence of cell-to-cell heterogeneities that are not considered in the model or reveal intrinsic noise due to molecular fluctuations (see discussion).

FGF and ephrin control the activation of the ERK pathway during early neural induction and could thus both govern the emergence of the neural cell fate. Because of the inverse relationship between the cell surface contact with FGF-expressing mesendoderm cells and that with ephrin-expressing ectoderm cells (Fig 1C), the a6.5>a6.7>a6.6>a6.8 profile of ERK

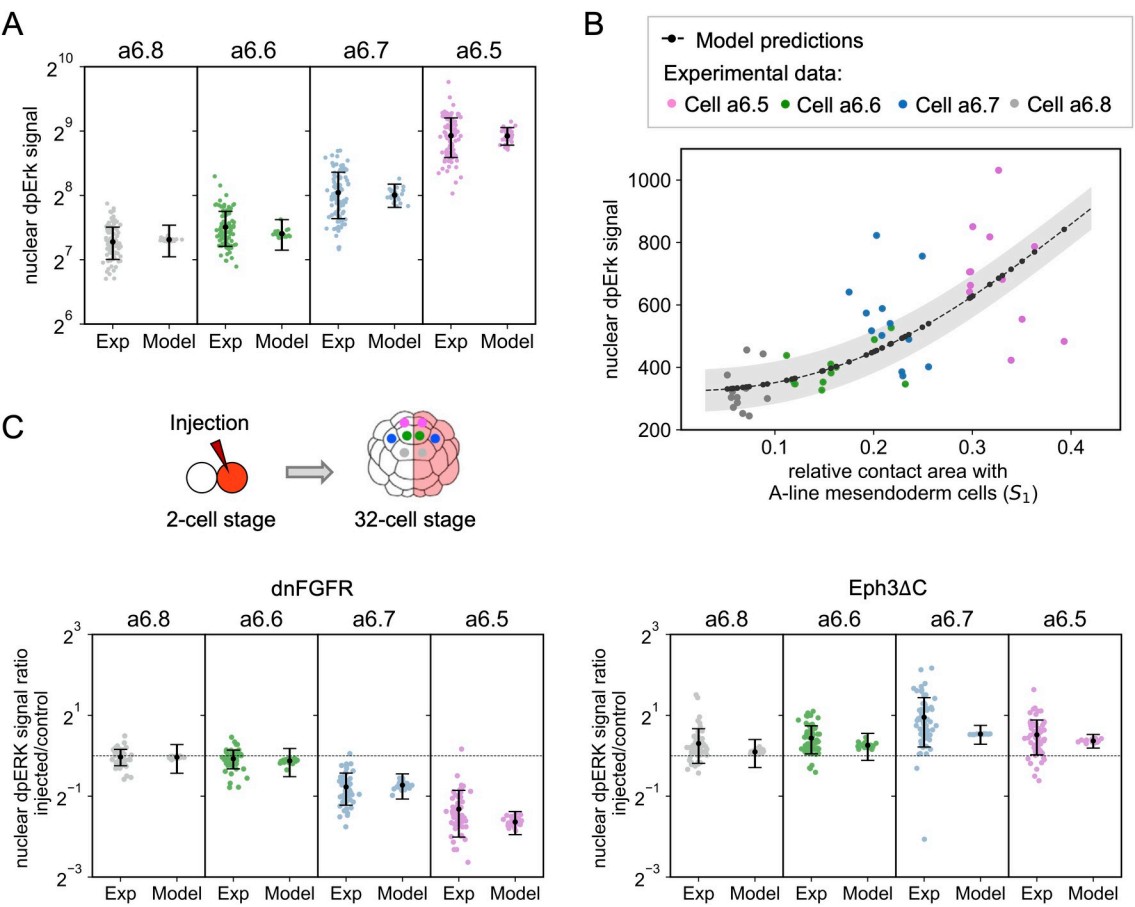

**Fig 2. Control of ERK activation by cell contact surfaces. (A)** Nuclear dpERK immunofluorescence (IF) signals in the a6.5, a6.6, a6.7 and a6.8 cell types as measured in IF experiments (left) and computed with the model (right, $Erk^f$ values). Each point represents a single cell and modeling results are computed using the measured values of $S_1$. Means and standard deviations are shown in black. A = 1850 and B = 155.11 in Eq (14). **(B)** Nuclear dpERK IF signals in individual a-line cells are shown as a function of the relative area of cell surface contact with A-line cells in experiments and in the model. Experimental data are shown in grey (a6.8 cell type), green (a6.6 cell type), blue (a6.7 cell type), magenta (a6.5 cell type), predictions of the model in black. Hill coefficient obtained by fitting the model predictions with a Hill function: 2.41. A = 3000 and B = 324.52 in Eq (14). The area shaded in grey represents the uncertainty on the model prediction. **(C)** Injected/control ratios of nuclear dpERK signal in dnFGFR half-injected embryos (on the left) and in Eph3ΔC half-injected embryos (on the right). Left and right columns show experimental dpERK IF and computed $Erk^f$ ratios, respectively. Injection of dnFGFR and Eph3ΔC were modeled by considering $R_T = 100$ and $Q_T = 10$, respectively. A = 1850 and B = 155.11 in Eq (14).

activation levels in the 32-cell stage ascidian embryo could rely either on a decreasing gradient of active SOS or on an increasing gradient of active p120RasGAP. The respective contributions of these opposite influences were investigated experimentally by several targeted manipulations of the two pathways and by modeling [8]. It was shown that the profile of ERK activation can be controlled by the gradual activation of FGF receptors, even in the absence of ephrin signals. Ephrin signaling is important to reduce the levels of ERK activation across all ectoderm cells. Modeling predicted a greater influence of SOS compared to p120RasGAP on ERK activation levels. In the following, we validated the model and the parameter values by simulating the various experiments conducted in [8].

The pharmacological inhibitor of ephrin receptors, NVP-BHG712 (henceforth NVP), has been shown to inhibit Eph signals in ascidian embryos [8,14]. When the model equations were solved with a minimal value of ephrin (0.001) to simulate the presence of the inhibitor, the

**Table 1. Default values of parameters.** All the units reported in the table are arbitrary. Parameter values were obtained by manual fitting to get best agreement with experimental observations. The values of the parameters of the two modules of the model were fitted separately. For the first module (model of ERK activity), fitting was performed manually to reproduce the experimental data obtained in ephrin inhibited embryos (S1B and S1C Fig), to set the values of the following parameters: $V_s$, $K_1$, $K_s$, $K_b$, $K_{erk}$, $K_d$, [FGF]. Fitting was then performed on the experimental data obtained in wild type embryos (Fig 2A and 2B) to set the values of the remaining parameters: $V_{rg}$, $K_2$, $K_{rg}$, $K_e$, [ephrin]. Finally, to fix the values of the parameters in the model of *Otx* expression ($k_{MM3}$, $K_{MM3}$, $v_{MM4}$, $K_{MM4}$, $k_{MM1}$, $K_{MM1}$, $v_{MM2}$, $K_{MM2}$, $v_b$, $v_o$, $K_a$, $K_i$, k) fitting was done on Fig 3D.

| Parameter | Definition | Value |
|---|---|---|
| A | Maximal level of dpERK immunofluorescence | Variable |
| B | Background level of dpERK immunofluorescence | Variable |
| C | Maximal number of smFISH Otx spots | Variable |
| D | Basal smFISH Otx spots count | Variable |
| [ephrin] | Concentration of ephrin | 5 ([concentration]) |
| [FGF] | Concentration of FGF | 5 ([concentration]) |
| k | Degradation rate for Otx | 0.2 ([1/time]) |
| $K_a$ | Half saturation constant for the activator Ets1/2 | 0.1 |
| $K_b$ | linear reaction rate of conversion from Ras-GTP into Ras-GDP due to contact surface independent RasGAP activity | 0.2 ([1/time]) |
| $K_d$ | Binding constant of FGF to its receptor FGFR | 25 ([concentration]) |
| $K_e$ | Binding constant of ephrin to its receptor Eph | 50 ([concentration]) |
| $K_{erk}$ | Fraction of Ras-GTP leading to half maximal ERK activation | 0.5 |
| $K_i$ | Dissociation constant for the repressor ERF2 | 0.1 |
| $k_{MM1}$ | Rate constant for the phosphorylation reaction of the repressor ERF2 | 12 ([1/time]) |
| $K_{MM1}$ | Michaelis-Menten constant for the phosphorylation of $I$ (ERF2) | 0.05 |
| $K_{MM2}$ | Michaelis-Menten constant for the dephosphorylation of $I_p$ (phosphorylated ERF) | 0.05 |
| $k_{MM3}$ | Rate constant for the phosphorylation reaction of the activator Ets1/2 | 12 ([1/time]) |
| $K_{MM3}$ | Michaelis-Menten constant for the phosphorylation of $A$ (Ets1/2) | 0.05 |
| $K_{MM4}$ | Michaelis-Menten constant for the dephosphorylation of $A_p$ (phosphorylated ERF2) | 0.05 |
| $K_{rg}$ | Number of ephrin-bound Eph receptors leading to half of the maximal p120RasGAP activation | 1200 |
| $K_s$ | Number of FGF-bound FGF receptors leading to half of the maximal SOS activation | 1200 |
| $K_1$ | Normalized half-saturation constant of SOS for its substrate Ras-GDP | 0.5 |
| $K_2$ | Normalized half-saturation constant of p120RasGAP for its substrate Ras-GTP | 0.2 |
| n | Hill coefficient for the ERK pathway | 2 |
| $Q_T$ | Total number of ephrin receptors present on the cell membrane | 2000 |
| $R_T$ | Total number of FGF receptors present on the cell membrane | 2000 |
| $S_1$ | Fraction of cell surface contact with FGF | Variable |
| $S_2$ | Fraction of cell surface contact with ephrin | Variable |
| $v_b$ | Basal Otx expression rate | 0.001 ([1/time]) |
| $v_{MM2}$ | Maximal rate at which $I_p$ (phosphorylated ERF2) is dephosphorylated | 1 ([1/time]) |
| $v_{MM4}$ | Maximal rate at which $A_p$ (phosphorylated Ets1/2) is dephosphorylated | 1 ([1/time]) |
| $v_o$ | $v_b+v_o$ is the maximal Otx expression rate | 1 ([1/time]) |
| $V_s$ | Maximal rate of SOS activation | 1 ([1/time]) |
| $V_{rg}$ | Maximal rate of p120RasGAP activation | 0.4 ([1/time]) |

a6.5>a6.7>a6.6>a6.8 profile of Erk* activation in the four cell types persists (S1C Fig), as observed experimentally. Moreover, the model still exhibits a positive correlation between the area of cell surface contact with FGF-expressing cells and ERK activation levels, although the dependence is smoother (S1B Fig, $n_H$ = 1.96). The effect of altering the SOS pathway was investigated experimentally by injecting a dominant negative form of the FGF receptor (dnFGFR) into one of the cells of the two-cell stage embryo. In these conditions, dnFGFR blocks FGF signals in one half of the embryo, while the non-injected side represents the control situation. The effect of dnFGFR was simulated by decreasing the number of FGF receptors

($R_T$) by a factor of 20 (i.e. $R_T = 100$). Observed ratios of dpERK IF signals between injected and control sides are shown in Fig 2C. Computations predict that the decrease in the number of FGF receptors leads to a marked decrease in the ERK activation levels in cell types a6.5 and a6.7: Erk* now ranges from ~$5 \cdot 10^{-6}$ in a6.8 cells to ~$5 \cdot 10^{-4}$ in a6.5 cells. Thus, the ERK pathway is practically totally inactive in all cell types when FGFR is inhibited. In a symmetrical manner, the model reproduces the effect of the injection of the dominant negative forms of the Eph3 receptor (Eph3ΔC) (Fig 2C). Best agreement with experimental data is found when dividing the number of ephrin/Eph receptors ($Q_T$) by 200 (i.e. $Q_T = 10$). Similarly, the effect of dominant negative p120RasGAP (RGΔGAP) was modelled by considering $V_{rg} = 0.01$ instead of 0.4 (S1D Fig).

In summary, simple phenomenological equations describing the regulation of the ERK signaling pathway by FGF receptor-controlled SOS activation and ephrin/Eph-controlled p120RasGAP activation can quantitatively account for experimental observations in the 32-cell stage ascidian embryo. Steady-state solutions of the model confirm that gradual ERK activation levels are controlled by the different cell surface contacts to FGF and ephrin-expressing cells. The relation between these surfaces and ERK activation levels is moderately nonlinear. The model predicts that the ERK pathway is far from being maximal at the 32-cell stage of the ascidian embryo, even in the most-strongly activated a6.5 cell type (S1A Fig).

## Dual control of the expression of the *Otx* gene by ERK

The *Otx* gene is a direct target of ERK signals during the initial step of ascidian neural induction [6,15]. Our previous results [8], modeled in the preceding section, have shown that ERK activation in the 32-cell stage ascidian embryo exhibits a relatively smooth profile in response to contact-dependent exposure to FGF signals (Figs 2A and 2B and S1B). In contrast, the transcription of the immediate-early gene *Otx* is bimodal, with a ON response that is restricted to neural precursors. We hypothesized that the ON/OFF response is, at least in part, due to the dual control of *Otx* expression [8]: Ets1/2 phosphorylation by ERK is required for its transcriptional activator activity [16], while phosphorylation of the ERF2 repressor by dpERK promotes its nuclear export, thus preventing its binding to DNA [12,13,17] (Fig 1A).

To further validate this hypothesis, we have extended the model of ERK activation studied in the previous section to include the relation between ERK activation, phosphorylated Ets1/2, unphosphorylated ERF2 and *Otx* expression. We first seek a single set of parameter values accounting for the observed levels of *Otx* expression in the four cell types. The existence of such a set would corroborate the assumption that the emergence of the neural cell fate in ascidian embryos is controlled by the contact areas with FGF and ephrin-expressing cells. Steady state levels of *Otx* smFISH spots computed with the model defined by Eqs (1–18), using the parameter values listed in Table 1, are shown in Fig 3A, together with experimental data for comparison (see also S3A Fig for the O values before the linear transformation into *Otx* smFISH spots). When simulating the inhibition of FGF signaling ($R_T = 100$ instead of 2000), O values drop to nearly zero in all cell types, which corresponds to the absence of *Otx* expression observed in dnFGFR-injected half embryos (Figs 3B and S4A). In the model as in the experiments, *Otx* expression is enhanced by inhibition of Eph signaling. This is noticed in the simulations of invalidation of the expression of ephrin/Eph receptors ($Q_T = 10$ instead of 2000) or of p120RasGAP enzymes ($V_{rg} = 0.01$ instead of 0.4), or by simulating the inhibition of the ephrin/Eph receptors by NVP ([ephrin] = 0.001 instead of 5) (see Figs 3B, 3C and S4). In all cases, these treatments reproduce the observed ectopic activation in a6.7 cells. This suggests that the moderate increase in ERK activation levels that results from the absence of Eph signaling (Figs 2C right and S1D) has a drastic effect on *Otx* expression in this cell type. In

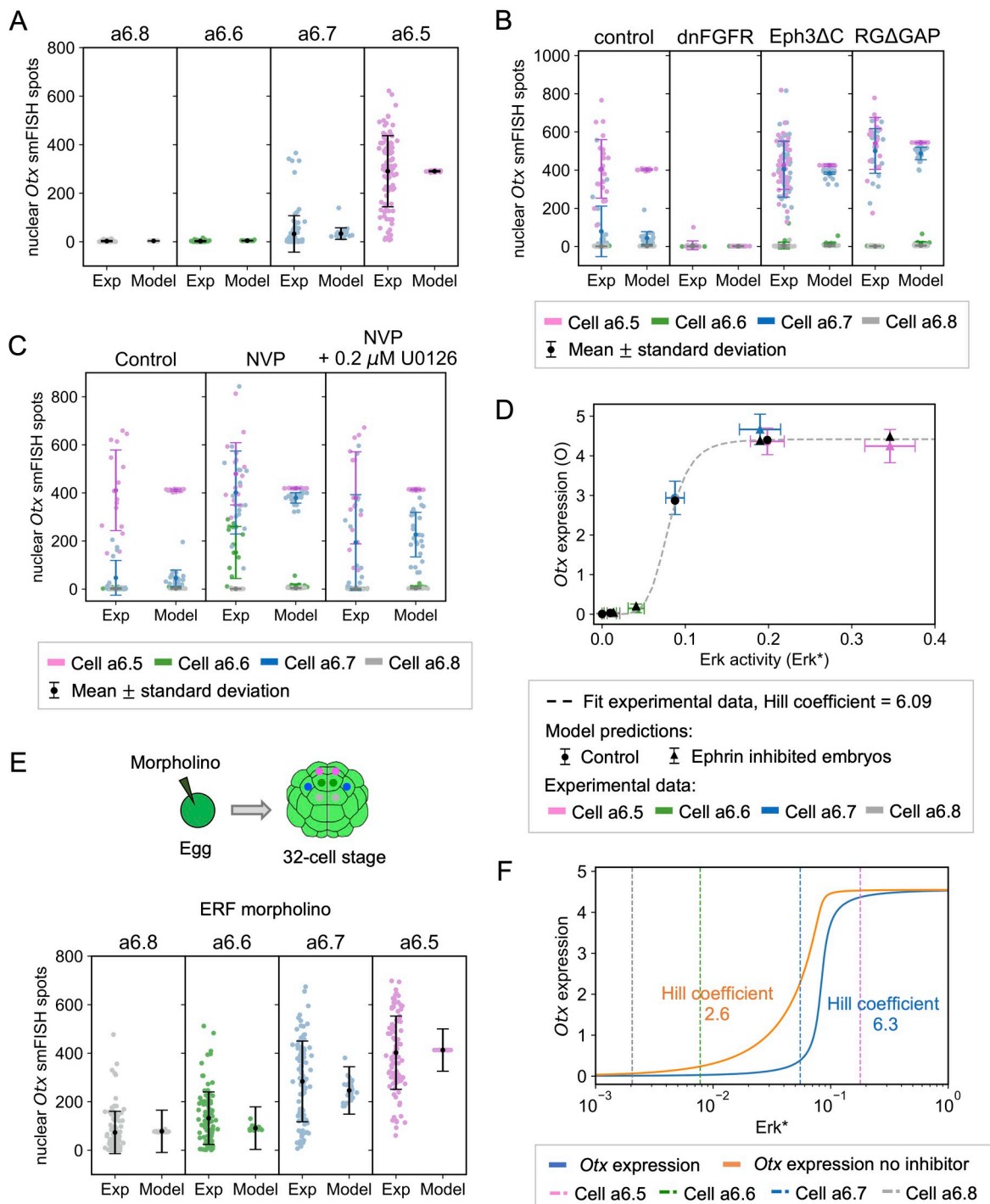

**Fig 3. Control of *Otx* expression by ERK. (A)** Levels of *Otx* expression in the a6.5, a6.6, a6.7 and a6.8 cell types as measured by single molecule fluorescence *in situ* hybridization (smFISH, left) and computed with the model (right, *Otx_smFISH*). Each point represents a single cell and modeling results are computed using the measured values of $S_1$. Means and standard deviations are shown in black. C = 66 and D = 2.75 in Eq (18). **(B)** *Otx* expression in control, dnFGFR, Eph3ΔC and RGΔGAP injected embryo halves. Left and right columns show experimental *Otx* smFISH spots and computed *Otx_smFISH*, respectively. Results for the four cell types are shown in the same columns with different colors. Each dot represents a single cell. Injection of dnFGFR, Eph3ΔC and RGΔGAP were modeled by considering $R_{tot}$ = 100, $Q_{tot}$ = 10, and $V_{rg}$ = 0.01, respectively. In Eq (18), C = 92 for the control embryos and for dnFGFR injected embryos, C = 95 for Eph3ΔC injected embryos and C = 122 for RGΔGAP injected embryos. D = 1.5 for the control and dnFGFR injected embryos, D = 2.71 for Eph3ΔC and D = 1.61 for RGΔGAP injected embryos. **(C)** Effect of the ephrin/Eph inhibitor NVP on *Otx* expression in the four cell types. On the left, control; in the middle, NVP-treated embryos; on the right, embryos treated with NVP and with the MEK inhibitor

U0126 0.2 $\mu$M. In the three cases, experimental *Otx* smFISH spots are shown on the left and computed $Otx_{smFISH}$ values on the right. Results for the four cell types are shown in the same column with different colors. Each dot represents a single cell. NVP and moderate U0126 treatment are simulated by considering [ephrin] = 0.001 and $K_{erk}$ = 0.6, respectively. In Eq (18), C = 94, D = 1.2. For B-C) the data for individual cell-types is shown in S4 Fig for clarity. **(D)** *Otx* expression as a function of ERK activation levels (Erk*) in the four cell types (colored points and triangles) and computed with the model ($Otx_{smFISH}$, black points and triangles). In the two cases, dots indicate the control while triangles indicate the ephrin-inhibited embryos. The experimental data were fitted by a Hill function, best-fit Hill coefficient = 6.09. For the experimental points, the value of Erk* corresponding to each cell was obtained by inversion of Eq (14), with A = 3200 and B = 0. The experimental value of O corresponding to each cell was obtained by inversion of Eq (18), with C = 120 and D = 0. Modelled *Otx* outputs were computed using the experimentally measured Erk* estimates as inputs. **(E)** Levels of *Otx* expression in the a6.5, a6.6, a6.7 and a6.8 cell types when unfertilized eggs have been injected with the ERF2-morpholino to prevent translation of ERF2. Injection of this morpholino is modeled by considering I as a constant equal to 0.01 in Eq (15). C = 75 and D = 73.4 in Eq (18). Modeling results for control embryos are the same as in panel (A). **(F)** *Otx* expression (O) as function of Erk*, computed with the model considering the presence (blue line, Hill coefficient = 6.3) or the absence (orange line, Hill coefficient = 2.6) of I (repressor). The Hill coefficients were computed using relation (25). Dashed vertical lines represents the mean values of Erk* for each cell type.

Eph-inhibited embryos, if ERK signaling is reduced by another means (low dose of the U0126 inhibitor of MEK that is simulated by increasing $K_{erk}$ (0.6 instead of 0.5)), an *Otx* output similar to the wild-type pattern is recovered (Figs 3C, right column and S4B).

In fact, a highly nonlinear relationship between *Otx* expression and ERK activation is observed in the experiments and in the model. In Fig 3D, values of Erk* and O corresponding to the experiments are inferred by inversion of Eq (14) and (18), respectively. Experimental values (colored dots and triangles) are in good agreement with model results (black dots and triangles) and can be fitted by a sharp Hill function (Hill coefficient = 6.09). For each cell type, inhibition of the ephrin pathway by the injection of Eph3ΔC shifts the representative points to the right. While the effect of this shift on *Otx* expression is negligible for the a6.5, a6.6 and a6.8 cell types that are quite far from the threshold, it moves the a6.7 from the steep part of the curve to the plateau. Thus, *Otx* expressions in the a6.7 and a6.5 cell types become similar. This threshold-like behavior involves a repressor of *Otx* expression, ERF2 [8]. The key role of this transcriptional repressor is visualized in Fig 3E–3F. Using the values of parameters calibrated on experimental data (Table 1), the levels of active, phosphorylated Ets1/2 activator (green curve in S3B Fig) and of active, unphosphorylated ERF2 repressor (red curve in S3B Fig) are plotted as a function of Erk*. These curves are sharp ($n_H$ = 6.3 for the two curves) despite the Michaelis-Menten kinetic functions, because the phosphorylation and dephosphorylation reactions of the two transcription factors are not far from saturation ($K_{MM1} = K_{MM2} = K_{MM3} = K_{MM4} = 0.05$), a situation close to zero-order ultrasensitivity [18], although different means of introducing ultrasensitivity between ERK and *Otx* are also compatible with our data (S7 and S8 Figs and see section 'Model predictions'). As a consequence of the combination of these two relatively sharp functions, the relation between *Otx* expression and ERK activation is characterized by a Hill coefficient of 6.3 (Fig 3F). Interestingly, when the effect of the repressor is not considered in the model (I = 0), the curve is not only shifted to the left because of the smaller effective binding constant of the activator, but also becomes much smoother (Hill coefficient = 2.6). This theoretical result is in line with the overall increase and smoother behavior of *Otx* expression in embryos injected with an ERF2 anti-sense morpholino oligo (compare Fig 3A and 3E). We thus conclude that the sharp relation between expression of the *Otx* gene and ERK activation is possible because the ERK pathway both promotes the expression of *Otx* and mitigates the repression of this expression.

## Sensitivity analysis

We next conducted a sensitivity analysis to assess the robustness of the model and to identify the parameters most affecting the model's behavior. Fig 4A shows the predicted levels of ERK activation in the different cell types when randomly changing the values of the parameters

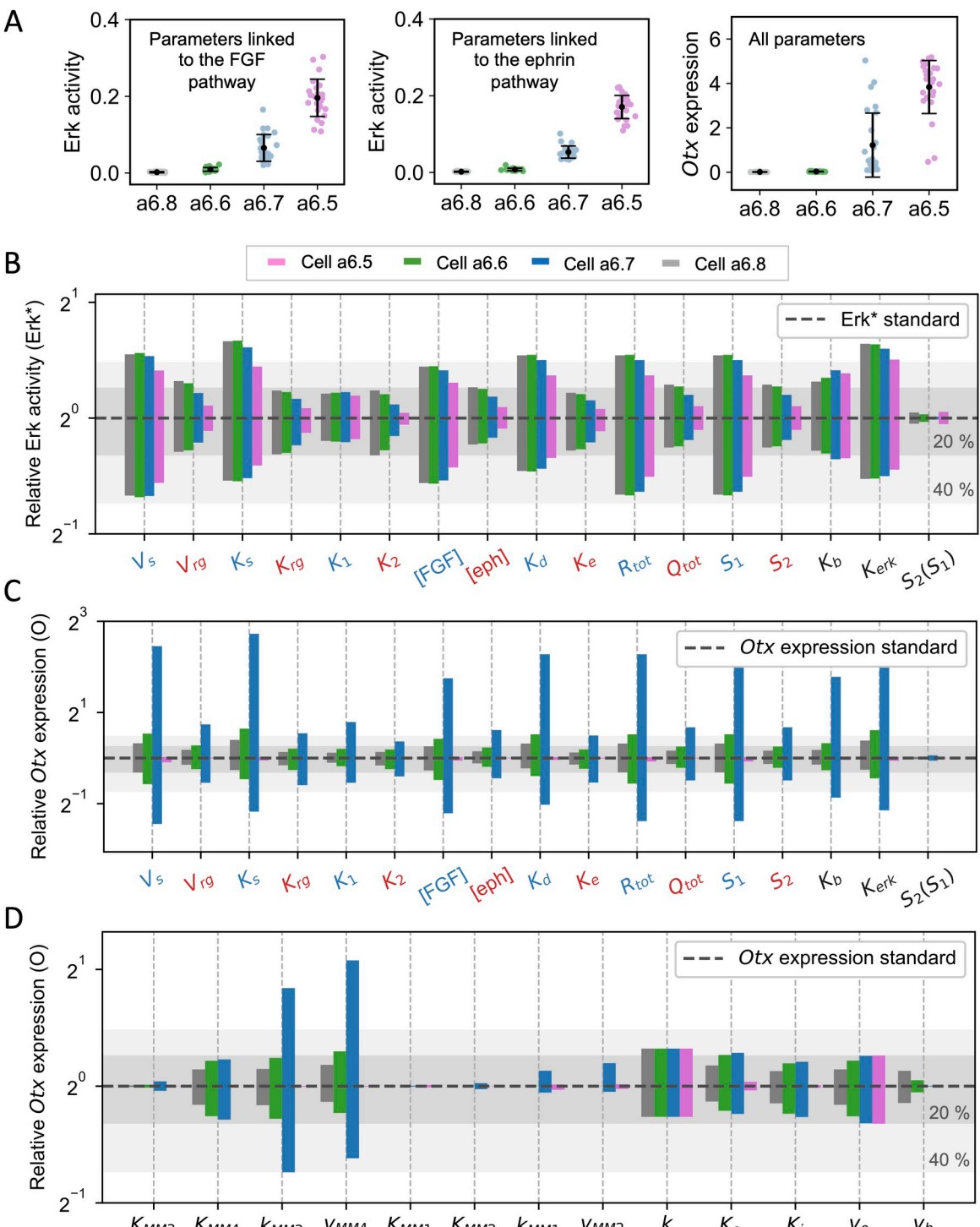

**Fig 4. Sensitivity analysis. (A)** On the left: levels of ERK activation (Erk*) in a6.5, a6.6, a6.7 and a6.8 cell types computed with the model when randomly varying the values of the parameters linked to the FGF pathway ([FGF], $K_d$, $K_1$, $K_s$, $V_s$, $R_T$) in the interval: standard value ± 20%. Middle: Erk* levels in the different cell types obtained when randomly varying the values of the parameters linked to the ephrin pathway ([ephrin], $K_e$, $K_2$, $K_{rg}$, $V_{rg}$, $Q_T$) in the interval: standard value ± 20%. Right: levels of *Otx* computed with the full model randomly varying the values of the parameters linked to the FGF and ephrin pathways, as above, as well as the other parameters (all the parameters in panel D plus $K_b$ and $K_{erk}$). **(B)** Erk* values computed when varying the value of one parameter at a time by ± 20%. Shown are the ratios between the computed values and the value obtained with the default values of parameters indicated in Table 1. Parameters affecting the SOS pathway are indicated in blue, while those affecting the p120RasGAP one are indicated in red. Results above $S_1$ and $S_2$

were obtained by considering $S_1$ and $S_2$ as separate inputs, thus disregarding Eq (12). These changes can also be regarded as changes in receptor densities (see Eq (6) and Eq (10)). Results for $S_2(S_1)$ were obtained by increasing/decreasing the value of the slope in Eq (12) by ± 20%, and then adjusting the intercept to best fit with experimental data. **(C)** O values computed when varying the value of one parameter at a time by ±20%. Results are obtained in the same way as for panel (B). **(D)** O values computed when varying the value of one parameter at a time by ± 20%. All panels show the ratios between the computed values and the value obtained with the default values of parameters indicated in Table 1 with $S_1$ set as the average value measured for each cell type. Shaded grey areas represent 20% and 40% changes in the levels of ERK activity/*Otx* expression obtained by changing the values of the parameters of the model.

linked to the FGF pathway (on the left) or the ephrin pathway (Fig 4A, middle) in the range defined by the standard value ± 20%. The spread in the level of ERK activation obtained in the first case is much larger than the spread obtained in the second one, suggesting that parameters linked to the FGF pathway have a higher influence on the level of ERK activation than parameters linked to the ephrin pathway. However, the average value remains practically unchanged in each cell. Along the same lines, changing all parameters ± 20% results in a robust profile of *Otx* expression (Figs 4A, left and S7F).

To investigate in more detail the sensitivity of the model to individual parameter change, the influence of varying the value of each parameter by ±20% on the steady states of ERK activity ($Erk^*$) and *Otx* expression (O) in the four cell types is presented in Fig 4B and 4C (see also Table 1 for the definition of each parameter). It is clear that ERK activation (Fig 4B) and *Otx* activation (Fig 4C) are more sensitive to changes in the values of parameters affecting the SOS pathway (indicated in blue) than the p120RasGAP one (indicated in red). This confirms and extends our previous conclusion that the control exerted by the FGF-receptor pathway is stronger than that exerted by ephrin/Eph [8]. The model is also quite sensitive to the value of $K_{erk}$ that represents the fraction of Ras-GTP leading to half-maximal activation of the ERK pathway. Given that this parameter controls the entire ERK pathway, which is modeled by a single Hill function Eq (1), it is intuitively expected that its value strongly affects the final value of $Erk^*$. Expression of *Otx* is dramatically affected in the a6.7 cell type when parameters promoting ERK activation are modified (Fig 4C), in agreement with the position of this cell on the ($Erk^*$, O) curve discussed in the previous section. Globally, the model is very robust to changes in the value of parameters describing the relation between $Erk^*$ and O (Fig 4D), except again for the a6.7 cells. For this cell type, values of O are strongly influenced by the rates of phosphorylation ($k_{MM3}$) and dephosphorylation ($V_{MM4}$) of the Ets1/2 activator. It should be noted, however, that this sensitivity analysis does not explore situations in which the relation between $Erk^*$ and *Otx* expression is not ultrasensitive since 20% changes around the standard $K_{MMi}$ values (0.05) still correspond to a zero-order regime. Most importantly, the overall a6.5>a6.7>a6.6>a6.8 profile of ERK activation and *Otx* expression is maintained in all tested parameter ranges (e.g Fig 4A). Thus, the mechanism of neural induction controlled by the cell contact area is robust against variations in the values of parameters.

## Model predictions

**Effect of ephrin-Eph signaling.** In this section, we use the model that has been calibrated on experimental data and shown to display robustness against changes in the values of parameters to make theoretical predictions. We investigated the sensitivity of ERK activation levels with respect to changes in the area of cell surface exposed to the ligands and to changes in ligand concentration in different conditions (in the presence or in the absence of ephrin). Because curves with a large steepness indicate a highly sensitive input-output relation, we used the Hill coefficient of the curves (see Modeling section for the definition) to measure sensitivity. The procedure is schematically represented in Fig 5A. In our previous study [8], we showed that ERK signaling is more sensitive to the cell surface contact to FGF-expressing cells

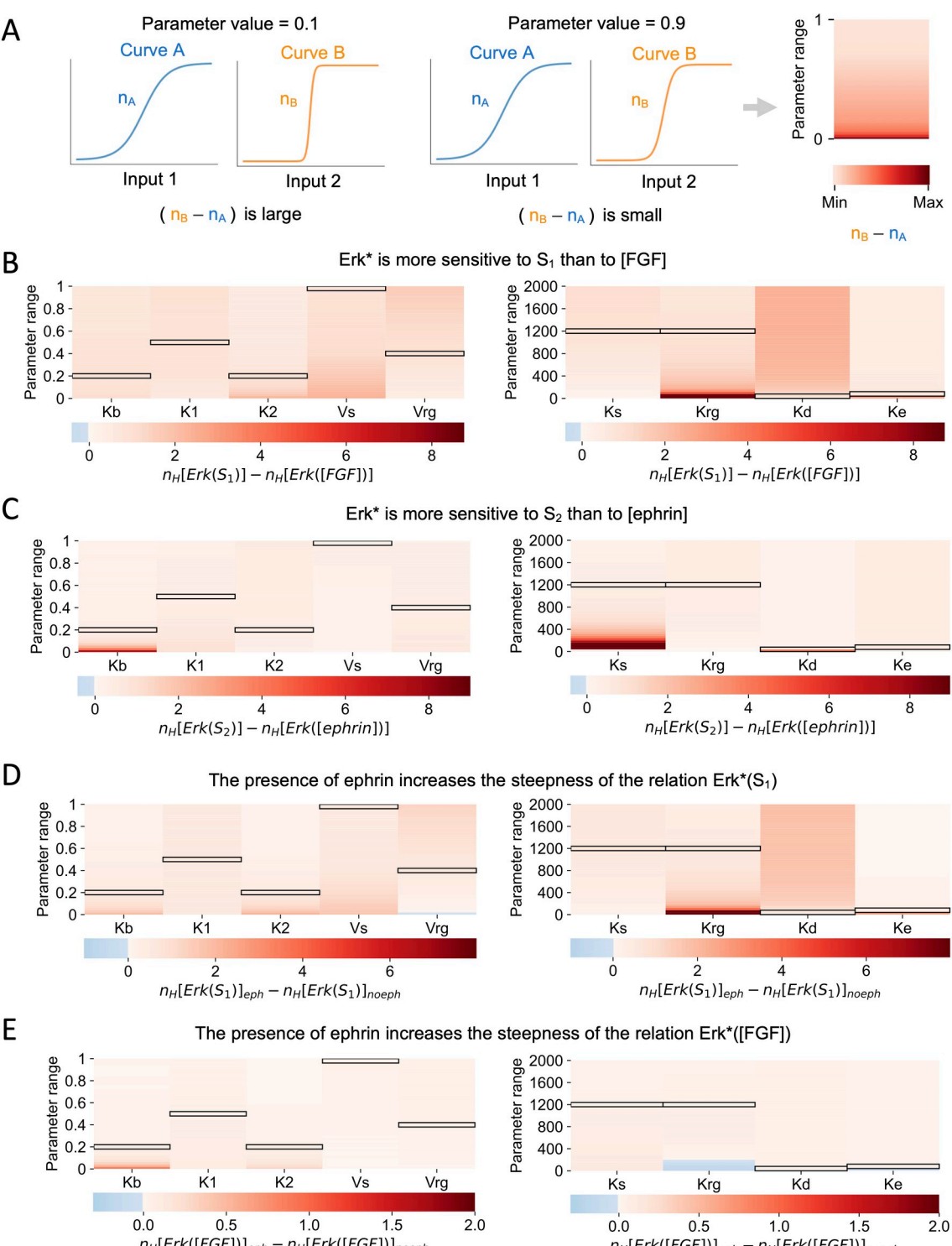

**Fig 5. Model predictions related to the factors affecting Erk activity. (A)** Graphical explanation of the numerical procedure. We consider two generic curves A and B with Hill coefficients $n_A$ and $n_B$ respectively. The shape of both curves depends on the value of a parameter p that can range from 0 to 1. On the left, the curves A and B are shown when the value of the parameter p is equal to 0.1. In this condition the difference of the two Hill coefficient $n_B$-$n_A$ is large. In the middle, the curves A and B are shown when the value of the parameter p is equal to 0.9. In this condition the difference of the two Hill coefficient $n_B$-$n_A$ is smaller. The right panel is the heatmap showing the differences between the Hill coefficients of curves A and B ($n_B$-$n_A$) computed for all values of p between 0 and 1. **(B)** Erk* values are more sensitive to $S_1$ than to [FGF]. Heatmaps show the difference between the Hill coefficient of the curve Erk*($S_1$)

and the Hill coefficient of the curve Erk*([FGF]). Hill coefficients obtained with default values of the parameters are 1.47 for Erk* ([FGF]) and 2.39 for Erk*($S_1$). To compute Erk*([FGF]) we fixed the value of $S_1 = 0.3$. In the two panels, $S_1$ and $S_2$ are related by Eq (12), which reflects the relation between the portions of cell surfaces exposed to FGF and to Ephrin, respectively. **(C)** Erk* values are more sensitive to $S_2$ than to [ephrin]. Heatmaps show the difference between the Hill coefficient of the curve Erk*($S_2$) and the Hill coefficient of the curve Erk*([ephrin]). Hill coefficients obtained with default values of the parameters are 1.17 for Erk*([ephrin]) and 1.57 for Erk*($S_2$). To compute Erk*([ephrin]) we fixed the value of $S_1 = 0.3$. In the two panels, $S_1$ and $S_2$ are related by Eq (12), which reflects the relation between the portions of cell surfaces exposed to FGF and to Ephrin, respectively. **(D)** The presence of ephrin increases the steepness of the relation between Erk* and $S_1$. Heatmaps show the difference between the Hill coefficient of the curve Erk*($S_1$) computed in the presence of ephrin and the Hill coefficient of the curve Erk*($S_1$) computed in the absence of ephrin ([ephrin] = 0.001). The Hill coefficient obtained with standard values of the parameters for Erk*($S_1$) in the absence of ephrin is 1.85. **(E)** The presence of ephrin increases the steepness of the relation between Erk* and [FGF]. Heatmaps show the difference between the Hill coefficient of the curve Erk*([FGF]) computed in the presence of ephrin and the Hill coefficient of the curve Erk*([FGF]) computed in the absence of ephrin ([ephrin] = 0.001). The Hill coefficient obtained with standard values of the parameters for Erk*([FGF]) in the absence of ephrin is 1.42. To compute Erk*([FGF]) we fixed the value of $S_1 = 0.3$. For all panels, the Hill coefficients were computed varying the values of the parameters from 0 to 1 ($K_b$, $K_1$, $K_2$, $V_s$, $V_{rg}$; left panels) or from 0 to 2000 ($K_s$, $K_{rg}$, $K_d$, $K_e$; right panels). Values of the Hill coefficients were obtained by curve fitting. The standard values of the parameters (Table 1) are highlighted with black boxes.

($S_1$) than to FGF concentration. Quantification of this statement, in terms of differences between Hill coefficients, is shown in Fig 5B keeping the relation between the area of each cell surface in contact with FGF expressing cells ($S_1$) and with ephrin expressing cells ($S_2$) observed in embryos (Fig 1C). For each column, we use the default values of all parameters (Table 1), except the one that is indicated. In the same line, we now investigate if ERK signaling is more sensitive to cell surface contact with ephrin-expressing cells ($S_2$) than to ephrin concentration itself. To this end, we computed the Hill coefficients of the Erk* vs $S_2$ relation and of Erk* vs [ephrin]. As shown in Fig 5C, ERK activation levels are more sensitive to cell surface contact to ephrin-expressing cells than to ephrin concentration.

The model can also be used to gain a deeper understanding of the role of ephrin, which was shown to prevent ectopic expression of *Otx* in the a6.7 cell by decreasing ERK activity below the threshold level in this cell type. Simulations reveal that the presence of ephrin also increases the steepness of the relation between Erk* and the cell surface contact to FGF-expressing cells, $S_1$ (Fig 5D). Interestingly, the steepness of the relation between Erk* and [FGF] is also increased by the presence of ephrin, although this effect is less pronounced (Fig 5E). In the two situations, the increase in steepness of the Erk* ([FGF]) (or Erk*($S_1$)) relation due to the presence of ephrin is maximal in the absence of ephrin-independent RasGAP activity ($K_b = 0$). The increased steepness of the Erk*([FGF]) relation is however not observed at low values of $K_{rg}$, which defines the sensitivity of p120RasGAP to ephrin-bound receptors (Fig 5E right). For small values of $K_{rg}$, p120RasGAP activation is actually insensitive to ephrin, which makes the computation physiologically meaningless. Results thus show that ephrin signaling reinforces the sensitivity of the cells to changes in cell surface contact with FGF-emitting cells.

**Comparison between FGF>FGFR >SOS and ephrin>Eph>RasGAP signaling.** As qualitatively discussed previously [8] and indicated above (Fig 4), the model predicts (Fig 6A) that ERK signaling is more sensitive to changes in the cell surface contact to FGF-expressing cells ($S_1$) than to the cell surface contact to ephrin-expressing cells ($S_2$) when considering these two surfaces as independent parameters.

Simulations shown in Fig 6B indicate that, in line with this property, Erk* is more sensitive to [FGF] than to [ephrin]. It is thus clear that there is an asymmetry in the respective influences of the FGF>FGFR>SOS and of the ephrin>Eph>RasGAP pathways in controlling the possible outcome of the neural fate. Simply stated, the upregulation of one of the pathways is not equivalent to the downregulation of the other. Interestingly, even in the absence of the ephrin-independent RasGAP activity ($K_b$), the Erk* vs [FGF] curve is slightly steeper than the Erk* vs [ephrin]. Indeed, Fig 6C indicates Hill coefficients of 2.13 and 1.80 respectively for the

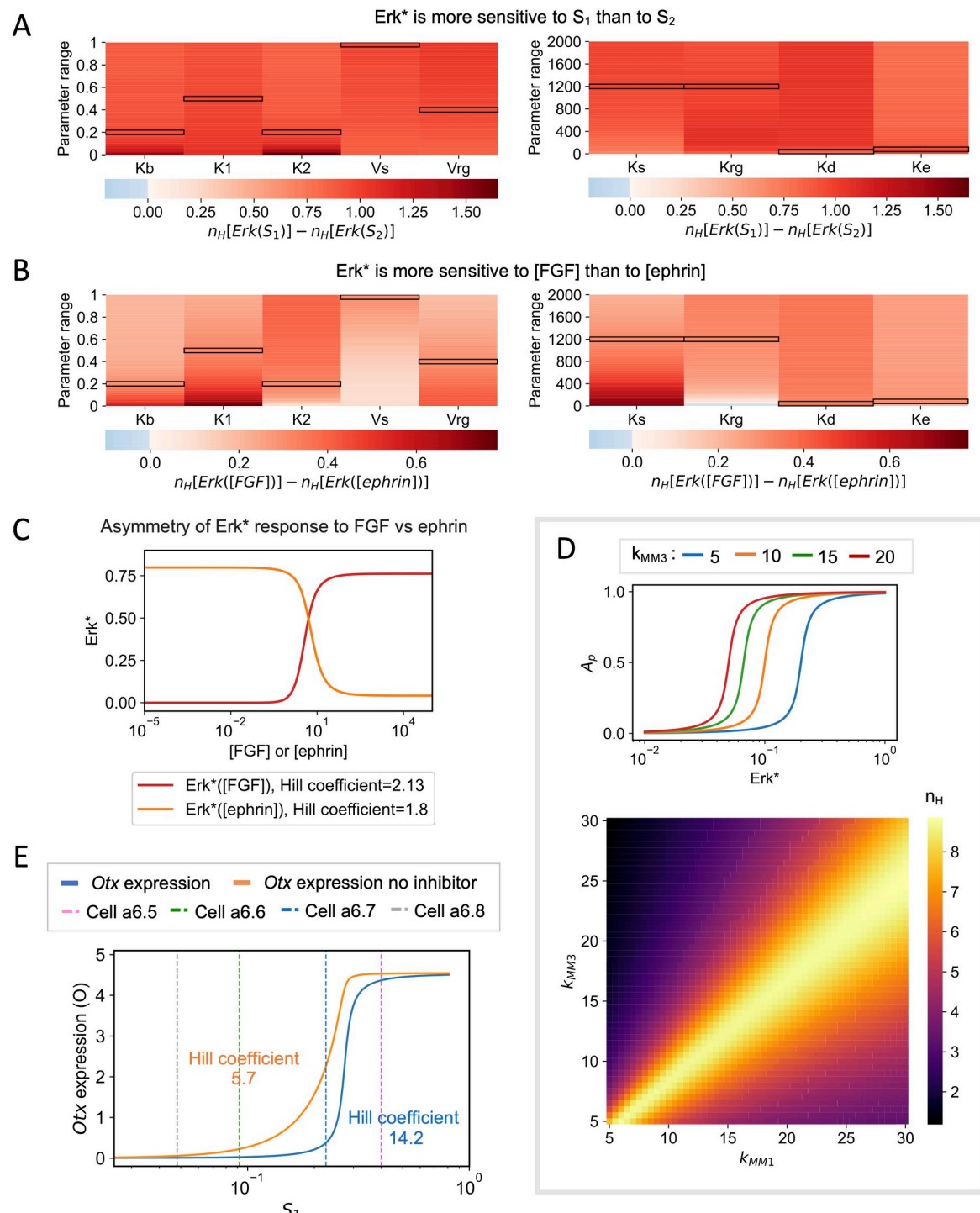

**Fig 6. Model prediction.** (A) Erk* values are more sensitive to $S_1$ than to $S_2$. Heatmaps show the difference between the Hill coefficient of the curve Erk*($S_1$) and the Hill coefficient of the curve Erk*($S_2$). Erk*($S_1$) was obtained by fixing the value of $S_2 = S^*$ ($S^* = 0.3$) and letting $S_1$ vary between 0 and 1-$S^*$. The same procedure was used to compute Erk*($S_2$). The Hill coefficients obtained with standard values of the parameters are 1.99 and 1.04 for Erk*($S_1$) and Erk*($S_2$), respectively. (B) Erk* values are more sensitive to [FGF] than to [ephrin]. Heatmaps show the difference between the Hill coefficient of the curve Erk*([FGF]) and the Hill coefficient of the curve Erk*([ephrin]). To compute Erk*([FGF]) and Erk*([ephrin]) we fixed the value of $S_1 = 0.3$. The standard values of the parameters are highlighted with black boxes. (C) ERK activation levels (Erk*) shown as a function of FGF concentration (in red) or ephrin concentration (in orange). Values of the parameters used: $V_s = V_{rg} = 1$, $K_1 = K_2 = 0.5$, $K_{rg} = K_s = 1200$, [FGF] or [ephrin] = 5, $K_d = K_e = 25$, $R_T = Q_T = 2000$, $S_1 = S_2 = 0.5$, $K_b = 10^{-6}$. Values of the Hill coefficients were obtained by curve fitting. (D) Upper panel: Effect of changing the $k_{MM3}$ in Eq (16) on

the relation between the fraction of phosphorylated activator, $A_p$ and active ERK, Erk*. Lower panel: Heatmaps showing the Hill coefficients of the relation between *Otx* and Erk* when changing the values of the $k_{MMi}$ in Eqs (16–17). **(E)** Relation between the level of *Otx* expression predicted by the model (O) and cell surface in contact with FGF-expressing mesendoderm cells ($S_1$) in the presence (blue curve, Hill coefficient = 14.2) or in the absence (orange curve, Hill coefficient = 5.7) of the repressor I. Dashed vertical lines represents the mean values of $S_1$ for each cell type. The Hill coefficients were computed using relation (26).

Erk* vs [FGF] and Erk* vs [ephrin] relations. For these curves, the values of all parameters have been taken the same for the FGF and the ephrin pathway, with $K_b \sim 0$ and $S_1$ and $S_2$ are both equal to 0.5. Even in these completely symmetrical conditions, the domain of values covered by the fraction of Ras-GTP (T) differs when varying FGF or ephrin concentrations (S5 Fig). In consequence, Erk* can never reach zero when increasing [ephrin] and can never reach 1 when increasing [FGF], which slightly modifies the steepness of the curves (Fig 6C).

**Effects of changing the kinetics of regulation of *Otx* expression.**    Finally, we explored the influence of the kinetics of the regulators of *Otx* expression on its switch-like behavior. It is highly sensitive to the steepness of the phosphorylation/ dephosphorylation cycle of the activator and the repressor, measured by the values of the $K_{MMi}$ (S6 Fig). Small values of these Michaelis-Menten constants are needed to get large Hill coefficients in the relation between *Otx* expression and ERK activation. These small values allow for zero-order ultrasensitivity [18]. The effects of the two regulators are however not symmetrical because the switch-like behavior cannot be obtained if only the repressor phosphorylation cycle has a steep dependence on ERK activity (low values of $K_{MM1}$ and $K_{MM2}$) (S6 Fig). Concerning the rates of phosphorylation, the switch-like behavior is optimized when phosphorylation rates for the repressor are slightly higher than the activator (Fig 6D). Cooperativities in the effects of the activators and repressors on *Otx* expression could play a similar role to zero-order kinetics (S7 Fig). A minimal enhancer of the *Otx* gene, named the a-element, mediates FGF responsiveness during ascidian neural induction and contains two ETS-binding sites in tandem, each of which is required for neural-specific expression in the ectoderm lineages [6]. When we consider cooperativity in Ets1/2 binding (see section 'Cooperativity in *Otx* activation'), the model reproduces experimental observations even for values of $K_{MMi}$ that correspond to a non-zero order regime (S7 Fig). Under these conditions, switch-like behavior of *Otx* is optimized when rates of phosphorylation are higher for the repressor, as above. In this scenario, the effects of the two regulators on the steepness of the *Otx* response are symmetrical (S8 Fig).

The relation between *Otx* expression (O) and the cell surface in contact with FGF-expressing mesendoderm cells ($S_1$) computed by the model is shown in Fig 6E. The Hill coefficients in the presence (14.2) and absence (5.7) of the repressor are about two times larger than those characterizing the relation between O and Erk* (6.3 and 2.6, respectively, see Fig 3F). Thus, the combination between a moderately nonlinear relation between Erk* and $S_1$ (Fig 2B) and a steep dependence of O on Erk* (Fig 3D), favored by the presence of the ERF2 repressor, allows for a quasi all-or-none dependence of the level of *Otx* expression on the cell surface contact. The relation between *Otx* expression and $S_1$, which would be difficult to obtain experimentally, also helps visualize how small variations in the values of the cell surface contact of a6.7 cells, which most probably occur in some embryos, would allow them to occasionally express *Otx*, in agreement with the experimental observations [7]. As well as slight changes in cell surface contact, it is also possible that cell-cell heterogeneity in parameter values from one embryo to another may contribute to the increased variability in *Otx* expression levels observed in a6.7 cells (Figs 4A, 4C, 4D and S7F). In short, the position of the a6.7 cell on the slope of the response curves make it more prone to variation in *Otx* expression compared to the other cells.

## Discussion

Modelling is much used in developmental biology to unravel the molecular mechanisms driving cell specification (see, for example: [19,20]). The model developed in this study supports the previously proposed hypothesis that the area of cell surface contacts are key determinants for ascidian embryogenesis [4]. This situation contrasts with the role of ligand concentration gradients which are widely employed in other model systems [1]. In the specific case of ascidian neural induction, a full mathematical description of the molecular mechanism, from activation of FGF and ephrin receptors to *Otx* expression, leads to a good quantitative agreement with *in vivo* observations, both in normal and perturbed conditions. These results, which display strong robustness with respect to the values of parameters, confirm that enough information is encoded in the area of cell surface contact with two signaling molecules to determine the epidermis *versus* neural fate induction in the ascidian 32-cell stage embryo. The model moreover reveals that the two signals do not play symmetrical roles. While FGFRs act as the primary determinant of ERK activation in each cell, differential ephrin/Eph receptor activation increases the sensitivity to FGF signaling. The somewhat secondary role of ephrin/Eph signaling is due to the existence of an ephrin-independent RasGAP activity. However, even in the absence of ephrin-independent RasGAP activity, there is an intrinsic asymmetry in the respective impacts of the Ras-GTP producing and reducing enzymes, as illustrated in Fig 6C. Finally, the larger role played by FGFR than by ephrin/Eph receptors in the control of ERK activity is also due to the small Ras-GTP/Ras-GDP ratio that is predicted by the model. As a consequence, SOS possesses a larger amount of available substrate, which makes changes in its level of activity more impactful on ERK activation levels than p120RasGAPs. As well as increasing the sensitivity of the response to FGF signals, ephrin/Eph signals also dampen ERK activation levels in all cells. This is critical in a6.7 cells in which ERK activation level is near the threshold for *Otx* expression.

The best agreement between modeling and experimental results is obtained for values of parameters leading to a predicted modest level of ERK activation in ectoderm cells. Such a modest level of activation results from the fact that extracellular [FGF] is much lower than the $K_d$ of FGF binding to its receptor, which means that the receptors are assumed to be far from saturation ([FGF] = Kd/5, see Table 1). In particular, modeled ERK activation level in the a6.5 cell under normal conditions is assumed to be lower than 25% of its maximum activation level (S1A Fig). This prediction is in line with the observation that nuclear dpERK IF signal levels detected in a6.5 cells can increase by a factor of ~2 in a6.5 cells of ephrin-inhibited embryos or by a factor of ~5 in maximally responding cells of ectodermal explants treated with high concentrations of exogenous FGF [8]. Thus, experimental observations support the model prediction that levels of ERK activation in individual cells is far from maximum.

Agreement between experimental observations and modelling predictions is obtained with a surprisingly simple mathematical description of ERK activation, in the form of a Hill function of the fraction of Ras-GTP (T). The adequacy of this description is further validated by the observation that the model also accounts for the measured levels of ERK activation in b-line ectoderm cells, taking into account values of their cell surface contacts with A-line and B-line mesendoderm cells. Simulation results agree with the restriction of ERK activation to b6.5 cells (S2 Fig) and thus also predict the high level of expression of *Otx* in these cells [15]. The adequacy of our simplified description of the ERK pathway is also corroborated by the fact that simulation results obtained with the Hill function can be recovered with a classical detailed model of the ERK activation pathway [21] using values of parameters lying within a reasonable physiological range (not shown).

In the model, sub-maximal ERK activation originates from the hypothesis that FGF concentration in the 32-cell stage ascidian embryo is smaller than the $K_D$ of FGF binding to its

receptor. This holds for all anterior ectoderm cells, because the FGF concentration is assumed to be the same for each cell. Differential levels of signaling between the cells is, in contrast, due to different areas of cell surface in contact with FGF-secreting mesendoderm cells. Assuming a homogenous distribution of FGF receptors, signaling regulated by the number of receptors appears to be more sensitive than signaling based on the modulation of ligand concentration (Fig 5B; [8]). That the number of possibly activatable receptors, rather than the agonist concentration, plays a primary role in the signaling pathway has been observed in other instances. For example, the existence and frequency of $Ca^{2+}$ oscillations in cells expressing the glutamate metabotropic receptor of type 5 (mGluR5) is controlled by the density of these receptors in the plasma membrane of stimulated cells rather than by the glutamate concentration [22]. In addition, during cellularization in early *Drosophila* embryos the activation of ERK in the ventral ectoderm in response to the EGFR ligand, Spitz, can also act as a sensor of the numbers of receptors available since halving the number of EGFRs results in reduction of ERK activation levels by approximately half [23]. In the invariantly cleaving early ascidian embryo, control by the surface-area determined number of receptors was predicted, in conjunction with binary induction outputs, to be sufficient to control specification in early embryos without additional layers of regulation [4].

Some assumptions of the model still require further investigation. The dispersion of the nuclear dpERK IF signals and *Otx* smFISH spots is much larger in the experiments than those predicted by the model. In the model, the dispersion of the points is assumed to have two qualitatively different origins. First, for each cell type, ERK signaling slightly differs from one cell to the other because cells slightly differ by their cell surface area in contact with FGF-expressing cells ($S_1$) and ephrin-expressing cells ($S_2$) (Fig 1C). Second, to compare the normalized values of ERK activation levels computed with the model to experimentally obtained nuclear dpERK IF signals, we took into account the background level of signal. The variation on this signal, which we equated to the variation in ERK activation levels in the less responding a6.8 cell, must thus also be considered (see the section "Estimation of the uncertainties" below). This explains why the error bars of the simulated values of the nuclear dpERK IF signals and nuclear *Otx* smFISH spots can exceed the dispersion of the individual points. However, in experimental measurements, if we consider a6.8 levels of ERK activation as background and normalize dpERK IF measurements in the remaining cells of each embryo half to the same side a6.8 value, the large variation in points persists (not shown). Thus, the variation in our experimental values does not only result from differential background levels of fluorescence. The larger dispersion of experimentally measured values suggests the existence of cell-to-cell differences (for example cell volumes and rate constants), other than relative cell surface areas, that are not considered in the model. In the same line, it is interesting to note that when fitting model outputs to experiments following inhibition of ERF2 function, a larger than usual background subtraction is required, indicating a larger than predicted basal level of *Otx* expression, even in cells with low ERK activity, when repression via ERF2 is lifted. Activation of *Otx* therefore likely involves factors in addition to ERF2/Ets1/2, with Gata.a [6,24,25] being the prime candidate currently under investigation. The presence of additional factors is not considered in the current model.

Another assumption of the model relates to the origin of the ultrasensitive relation between ERK and *Otx* expression. While we have shown that the presence of the ERF2 repressor further increases the steepness of the relation, the already non-linear relation between ERK and *Otx* expression ($n_H$ = 2.6, see Fig 3F) was postulated to arise from the presence of zero-order ultrasensitivity in the relationship between Ets1/2 and ERK* [18]. There is actually no experimental evidence for such phenomenon in the regulation of *Otx* expression, although it has been reported to be the case for other developmental-related processes [26]. Other possibilities,

such as cooperative binding of the effectors (as shown in S7 Fig), could be involved and play the same role as zero-order ultrasensitivity from a mechanistic point of view, without changing the general conclusions drawn from the present study. Further investigation of the mechanism by which a graded ERK signal is converted into a bimodal transcriptional output of immediate-early genes during ascidian neural induction would improve our general knowledge of the input-output relationship of the numerous ERK-dependent processes.

## Model

We developed a computational model to describe the regulation of *Otx* expression by the FGF- and ephrin-regulated ERK pathway during ascidian neural induction. The model results from the combination and parametrization of the three independent modules that were developed in our previous work to help understand how signaling inputs are integrated to activate ERK signaling on one hand, and how ERK signaling modulates *Otx* expression on the other hand [8]. Here we combine these modules and calibrate the values of the parameters to compare the outcome of the full model with experimental results.

### Model of ERK activation

We assumed that FGF and ephrin act as short-range ligands, with ephrins being membrane-tethered ligands, and that the concentration of extracellular FGF ([FGF]) between mesendoderm and ectoderm cells is uniform. This hypothesis is motivated by the observation that all a-line ectoderm cells are in contact with A-line mesendoderm cells that express uniform levels of FGF (see S1A Fig of [8]). Similarly, we assumed the concentration of extracellular ephrin ([ephrin]) to be constant. In line with experimental observations [8], we considered that the density of FGF receptors on the plasma membrane of ectoderm cells is uniform. In the absence of experimental data, we similarly considered the density of Eph receptors on the plasma membrane of ectoderm cells to be uniform. Thus, the number of receptors in contact with the ligand is different for each ectoderm cell, since it is proportional to the area of cell surface in contact with FGF- or ephrin- expressing cells.

The activation (phosphorylation) of the extracellular signal-regulated kinase, ERK, is induced by Ras-GTP (see Fig 1A) via a cascade of phosphorylations that is not considered explicitly. Assuming that the total amount of Ras-GDP and Ras-GTP is conserved, we described the relation between the fraction of active, doubly phosphorylated ERK ($Erk^*$) and the fraction of Ras bound to GTP ($T$) as a Hill function, i.e.:

$$Erk^* = \frac{Erk^A}{Erk_{\max}} = \frac{T^n}{T^n + K_{erk}^n} \tag{1}$$

$$where \quad T = \frac{[Ras - GTP]}{[Ras - GTP] + [Ras - GDP]}$$

$Erk_{\max}$ is the maximal level of ERK activation ($Erk^A$), $n$ is the Hill coefficient and $K_{erk}$ is the fraction of Ras-GTP leading to half maximal ERK activation.

The evolution equation for the fraction of Ras-GTP is (see Fig 1A):

$$\frac{dT}{dt} = V_{SOS}([FGF]) - V_{\mathrm{p120RasGAP}}([ephrin]) - V_{RasGAP}^b \tag{2}$$

where $V_{SOS}([FGF])$ is the conversion rate from Ras-GDP to Ras-GTP (mediated by the FGF-dependent SOS activity), $V_{\mathrm{p120RasGAP}}([ephrin])$ is the conversion rate from Ras-GTP to Ras-GDP (mediated by the ephrin-dependent p120RasGAP activity) and $V_{RasGAP}^b$ is the conversion

rate from Ras-GTP to Ras-GDP mediated by ephrin-independent RasGAP activity. We will now describe these three processes.

**SOS-mediated conversion from Ras-GDP to Ras-GTP.** Ras-GDP is transformed into Ras-GTP by SOS following Michaelian kinetics. Thus, the conversion rate $V_{SOS}$ can be written as:

$$V_{SOS} = V_1 \frac{(1 - T)}{K_1 + 1 - T} \tag{3}$$

where $V_1$ is the normalized maximal rate of conversion of Ras-GDP to Ras-GTP by SOS and $K_1$ is the normalized half-saturation constants of SOS for its substrate Ras-GDP. To take the cell-contact dependent number of ligand-bound FGF receptors into account, we considered that SOS must be in its active state SOS* (namely membrane-recruited) to promote the formation of Ras-GTP. The reaction for the activation of SOS depends on the number of FGF-bound receptors, $R_b$. So, we re-write the rate $V_1$ as:

$$V_1 = V_S \frac{R_b}{K_S + R_b} \tag{4}$$

where $V_S$ is the maximal rate of SOS activation and $K_S$ is the number of FGF-bound receptors leading to half of the maximal SOS activity. We do not model explicitly the heparan-sulfate dependent dimerization of bound FGF receptors that precedes activation of SOS. $R_b$ is given by:

$$R_b = R \frac{[FGF]}{[FGF] + K_d} \tag{5}$$

where $[FGF]$ is the extracellular concentration of FGF, $R$ is the number of receptors possibly in contact with FGF, and $K_d$ is the binding constant of FGF to its receptor. Since we assumed that the density of FGF receptors on the ectoderm cell membrane is constant, the number of receptors in contact with FGF can be written as:

$$R = R_T S_1 \tag{6}$$

where $R_T$ is the total number of FGF receptors present on a cell, and $S_1$ is the fraction of cell surface exposed to FGF. In the absence of experimental data for ascidian embryos, we choose $R_T = 2000$, in line with values reported for mammalian cells [27].

**Ephrin dependent RasGAP-mediated conversion from Ras-GTP to Ras-GDP.** The conversion of Ras-GTP to Ras-GDP is mediated by p120RasGAP following Michaelian kinetics. Thus, the conversion rate $V_{Ras-GAP}$ can be written as:

$$V_{Ras-GAP} = V_2 \frac{T}{K_2 + T} \tag{7}$$

where $V_2$ is the normalized maximal rate of conversion of Ras-GTP to Ras-GDP by p120Ras-GAP and $K_2$ is the normalized half-saturation constant of p120RasGAP for its substrate Ras-GTP.

To take the cell-dependent number of Eph receptors into account, we considered that p120RasGAP must be in an active state p120RasGAP* (membrane-recruited) in order to promote the formation of Ras-GDP.

The activation of p120RasGAP depends on the number of receptors bound to ephrin ($Q_b$). Thus, we re-write the transformation rate $V_2$ as:

$$V_2 = V_{rg} \frac{Q_b}{K_{rg} + Q_b} \tag{8}$$

$$\text{where} \quad Q_b = Q \frac{[ephrin]}{[ephrin] + K_e} \tag{9}$$

$V_{rg}$ is the maximal rate of p120RasGAP activation, $K_{rg}$ is the number of ephrin-bound Eph receptors leading to half of the maximal p120RasGAP activation. $Q$ is the number of Eph receptors in contact with ephrin, $[ephrin]$ is the ephrin concentration and $K_e$ is the binding constant of ephrin to its receptor.

Since we assumed that the density of Eph receptors on the ectoderm cell membrane is constant, the number of receptors in contact with ephrin can be written as:

$$Q = Q_T S_2 \tag{10}$$

where $Q_T$ is the total number of Eph receptors present on a cell, and $S_2$ is the fraction of cell surface exposed to ephrin.

**Ephrin independent RasGAP-mediated conversion from Ras-GTP to Ras-GDP.** The presence of basal GAP activity was considered since transcriptome datasets [28] indicate the presence of at least three further RasGAPs in the early ascidian embryo (*IQGAP1/2/3*, *Neurofibromin*, and *RASA2/3*). The conversion from Ras-GTP into Ras-GDP due to contact surface independent RasGAP activity is modeled by a linear reaction rate $K_b$. Thus, the conversion rate $V_{RasGAP}^b$ can be written as:

$$V_{RasGAP}^b = K_b T \tag{11}$$

**Empirical relation between fractions of cell surfaces in contact with FGF and with ephrin.** To account for the fact that the number of Eph receptors exposed to ephrin changes together with the number of FGF receptors exposed to FGF, i.e., that cells that have the highest number of FGF receptors in contact with FGF (R), have also the lowest number of Eph receptors in contact with ephrin ($Q$), we used the experimental data from Fig 1C. Plotting $S_2$ as a function of $S_1$ and fitting the experimental data with a linear function, we obtained the following relation between $S_1$ and $S_2$:

$$S_2 = -1.13\, S_1 + 0.91 \tag{12}$$

Everything considered, the evolution equation for $T$ (Eq (2)) becomes:

$$\frac{dT}{dt} = V_s \frac{R_b}{K_S + R_b} \frac{(1 - T)}{K_1 + 1 - T} - V_{rg} \frac{Q_b}{K_{rg} + Q_b} \frac{T}{K_2 + T} - K_b T \tag{13}$$

We solved the equation for $T$ at equilibrium, to get $T$ as a function of $[FGF]$ or as a function of $S_1$. We then computed the fraction of active ERK using Eq (1).

Usually, the quantity measured experimentally is the level of anti-dpERK IF-signals. Thus, to compare the simulation results with experiments we considered that the level of ERK

fluorescence ($Erk^f$) is a linear function of ERK activation level ($Erk^*$):

$$Erk^f = A \cdot Erk^* + B \tag{14}$$

Where $A$ is the maximal ERK anti-dpERK IF signal and $B$ is the background dpERK IF signal. Taking into account that $Erk^*$ computed with the model was close to zero in the a6.8 cell type, we chose B as the average level of fluorescence in the a6.8 cell type in each experiment. A was defined empirically in order to obtain the best fit to the data.

We have verified (not shown) that our theoretical predictions remain unchanged when using a detailed model to describe the activation of the ERK pathway [21].

## Model of *Otx* expression

*Otx* expression is regulated by dpERK (Fig 1A). A minimal enhancer of the *Otx* gene, named the *Otx* a-element, contains three GATA and two ETS (E26 transformation-specific)-binding sites required for neural-specific expression in the ectoderm lineages via Gata.a and Ets1/2 transcription factors [6,24].

The Ets2 repressor factor 2 (ERF2) [29] recognizes the same binding site as the transcriptional activator Ets1/2 and its activity is negatively controlled by ERK via nuclear export [12,13,17]. Thus, we considered that *Otx* expression is enhanced by the stimulation of an activator (A; e.g. Ets1/2) and is inhibited by a repressor (I; e.g. ERF2). The phosphorylation of both the activator and the repressor is controlled by ERK. Thus, we modeled *Otx* expression with the following evolution equation:

$$\frac{dO}{dt} = v_b + v_o \frac{A_p}{K_a\left(1 + \frac{I}{K_i}\right) + A_p} - kO \tag{15}$$

where $v_b$ is the basal *Otx* expression rate, $(v_b + v_o)$ is the maximal *Otx* expression rate and $k$ is the degradation rate for *Otx*. $K_a$ is the half saturation constant for the activator $A_p$ while $K_i$ is the dissociation constant for the repressor. In Eq (15), it is considered that ERF2 ($I$) is a competitive inhibitor of Ets1/2 ($A$). Thus, there is an effective half-saturation constant that depends on the concentration of $I$, which is equal to $K_a\left(1 + \frac{I}{K_i}\right)$. When $I$ is large, this effective constant is much larger than $K_a$ and, when $I$ is small, this effective constant approaches $K_a$.

The evolution equations describing the phosphorylation and dephosphorylation of Ets1/2 and ERF2 are:

$$\frac{dA_p}{dt} = k_{MM3} Erk^* \frac{(1 - A_p)}{K_{MM3} + 1 - A_p} - v_{MM4} \frac{A_p}{K_{MM4} + A_p} \qquad A \leftrightarrow A_p \tag{16}$$

$$\frac{dI}{dt} = v_{MM2} \frac{(1 - I)}{K_{MM2} + 1 - I} - k_{MM1} Erk^* \frac{I}{K_{MM1} + I} \qquad I \leftrightarrow I_p \tag{17}$$

where $A_p$ represents the fraction of active, phosphorylated activator; $(1-A_p)$ the fraction of inactive, unphosphorylated activator; I the fraction of active, unphosphorylated repressor and $(1-I)$ the fraction of inactive, phosphorylated repressor. For the two phosphorylation reactions, the rate constants $k_{MM3}$ and $k_{MM1}$ of A and I phosphorylation are multiplied by the fraction of doubly phosphorylated ERK ($Erk^*$). $v_{MM4}$ and $v_{MM2}$ are the maximal rates at which $A_p$ and $I_p$ are dephosphorylated. $K_{MM3}$ and $K_{MM1}$ are Michaelis-Menten constants for the phosphorylation of A and I, respectively, and $K_{MM2}$ and $K_{MM4}$ are Michaelis-Menten constants for the dephosphorylation of $I_p$ and $A_p$, respectively. $K_{MM3}$ and $K_{MM4}$ are normalized with respect

to the total activator concentration, while $K_{MM2}$ and $K_{MM1}$ are normalized with respect to the total repressor concentration.

Since the quantity measured experimentally is the counts of smFISH *Otx* spots, to compare the simulations results with experiments we considered that the level of counts of smFISH *Otx* spots ($Otx_{smFISH}$) is a linear function of the level of *Otx* expression ($O$):

$$Otx_{smFISH} = CO + D \tag{18}$$

where $C$ is the maximal number of smFISH *Otx* spots and $D$ is the basal smFISH *Otx* spot count. Taking into account that the value of $O$ computed with the model was close to zero in the a6.8 cell type, we chose $D$ as the *Otx* smFISH count in the a6.8 cell type in each experiment. C was defined empirically in order to obtain the best fit to the data.

To obtain an expression for *Otx* expression ($O$ and $Otx_{smFISH}$) as a function of the inputs of the signaling cascade ($S_1$) we combined the models of ERK activity and of *Otx* expression presented in the previous sections. We computed ERK activity using Eq (1) and we used this as the input (Erk*) to compute the fraction of active Ets1/2 ($A_p$) and active ERF2 ($I$) using Eqs (16) and (17). Then we computed the level of *Otx* expression by solving Eq (15). All equations are solved at steady state.

Equations are solved analytically at steady-state and solutions evaluated for all investigated sets of values of parameters with codes written in Python. All the codes and the experimental data are available at https://github.com/rossanabettoni/Model-of-neural-induction-in-the-ascidian-embryo.

## Model of ERK activation in b-line cells

b-line ectoderm cells are in contact both with A-line mesendoderm cells that express a uniform level of FGF and with B-line mesendoderm cells (B6.1, B6.2, B6.3 and B6.4) that express different levels of FGF (S2 Fig) [8]. Therefore, Eqs (5) and (6) do not appropriately describe activation of the FGF pathway in these cells. To model activation of the ERK pathway in the b-line cells, we considered two distinct populations of FGF receptors. The first population encompasses the receptors located on the surface in contact with the B-line cells. For each b-line cell, the number of FGF-bound receptors at the interface with cell B6.i (with i ranging from 1 to 4) is given by:

$$R_{bB6.i} = R_T S_{1B6.i} \frac{[FGF]_{B6.i}}{[FGF]_{B6.i} + K_d} \tag{19}$$

where $S_{1B6.i}$ is the area of the b-line cell surface in contact with cell B6.i and $[FGF]_{B6.i}$ is the modelled level of extracellular FGF concentration at this interface.

The second population of FGF receptors encompasses those located on the surface portion in contact with A-line mesendoderm cells. On this surface portion, the number of FGF-bound receptors is given by Eq (5). Thus, the total number of bound FGF receptor for a given b-line cell is:

$$R_b = R_T S_1 \frac{[FGF]}{FGF + K_d} + \sum_{i=1}^{4} R_T S_{1B6.i} \frac{[FGF]_{B6.i}}{[FGF]B_{6.i} + K_d} \tag{20}$$

Values of $S_1$ and $S_{1B6.i}$ are given on github (see above). The level of extracellular [FGF] at each interface was estimated from the experimentally measured gene expression levels of the *FGF9/16/20* ligand (see S1A in [8]). As these data only allow comparison with respect to the concentration of FGF perceived by the A-line mesendoderm cells for which [FGF] was taken equal to 5 (Table 1), we consider $[FGF]_{B6.1} = 5$, $[FGF]_{B6.2} = 0.7$, $[FGF]_{B6.3} = 0.08$ and $[FGF]_{B6.4}$

= 0.7. For this analysis, we disregarded Eq (12) and entered $S_1$ and $S_2$ measurements as independent variables. Experimental measurements for $S_1$ and $S_{1B6.i}$ and $S_2$ are indicated in S2A Fig.

## Cooperativity in *Otx* activation

Cooperativity in the Ets1/2-dependent *Otx* activation could also contribute to the ultrasensitivity in the *Otx* response to ERK activation. Indeed, the *Otx* a-element contains two ETS-binding sites [6].

We modelled cooperativity in Ets1/2 binding introducing a Hill coefficient equal to 2 in the evolution equation for *Otx*:

$$\frac{dO}{dt} = v_b + v_o \frac{A_p^2}{K_a^2 \left(1 + \frac{I}{K_i}\right)^2 + A^2{}_p} - kO \tag{21}$$

Eq (21) and the following parameters ($K_{MM1} = K_{MM2} = K_{MM3} = K_{MM4} = 0.3$, $k_{MM1} = k_{MM3} = 14$) were used to obtain the results presented in S7 Fig. This figure illustrates that zero-order ultrasensitivity in Eqs (16) and (17) can be replaced by cooperativity in Eq (15).

## Experimental methods

All experimental results, except the data obtained in b-line cells, are taken from our previous work [8]. Experimental measurements of cell-surface contacts and dpERK IF levels in b-line cells were obtained as described previously [8].

## Estimation of uncertainties

**Anti-dpERK immunofluorescence.** Given a complete set of parameter values including $S_1$ and $S_2$, there is no uncertainty on the deterministically modeled value of ERK activity ($Erk^*$). However, to compare with experimental data, we computed the average of 25 values of $Erk^*$ ($Erk_M$), corresponding to 25 couples of ($S_1$, $S_2$) values of individual cells. The uncertainty on this computed average activity ($\Delta Erk_M$) is the standard deviation.

Experimentally, ERK activation levels are quantified by immunofluorescence (IF) signals. As described above (Eq (14)), we assumed that IF signals are a linear function of $Erk^*$:

$$Erk^f = A \cdot Erk^* + B$$

where B is the background value of IF signals.

In the absence of other information, we postulated that the uncertainty on B ($\Delta B$) is the same as the standard deviation of the experimental values of dpERK IF signals measured in the a6.8 cells. Although it may be unlikely that there is zero variability between the true value of ERK activation level and the measured fluorescence intensity in terms of slope, we have no means to measure or estimate this. Therefore, we considered that there is no uncertainty on A.

Thus, the uncertainty $\Delta Erk^f$ is equal to $\Delta B$ for each cell, i.e. for each couple of ($S_1$, $S_2$) values. In Fig 2A, S1C and S2B the uncertainties $\Delta Erk^f$ for the single cells are not shown, for clarity reasons. The average ERK fluorescence for a given cell type is computed as:

$$Erk_M{}^f = A \cdot Erk_M^* + B$$

with the corresponding uncertainty:

$$\Delta Erk_M{}^f = \sqrt{\left(A \cdot \Delta Erk_M^*\right)^2 + \Delta B^2} \tag{22}$$

This uncertainty is indicated on the figure panels as the model error bars.

## Ratios between anti-dpERK immunofluorescence levels in treated and non-treated cells (for Figs 2C and S1C)

The ratio between the level of dpERK IF signals in the injected side of the embryo and the level of dpERK IF signals in the control side of the embryo (for a single cell) is:

$$r = \frac{Erk_{injected}^{f}}{Erk_{control}^{f}}$$

With uncertainty:

$$\frac{\Delta r}{r} = \sqrt{\left(\frac{\Delta Erk_{injected}^{f}}{Erk_{injected}^{f}}\right)^2 + \left(\frac{\Delta Erk_{control}^{f}}{Erk_{control}^{f}}\right)^2}$$

where $\Delta Erk_{injected}^{f}$ and $\Delta Erk_{control}^{f}$ are equal to $\Delta B$. In Figs 2C and S1D the uncertainties $\Delta r$ for the single cells are not shown.

The average of the ratios for each cell type was computed as:

$$r_M = \frac{Erk_M^{f}(inj)}{Erk_M^{f}(control)}$$

With corresponding uncertainty:

$$\frac{\Delta r_M}{r_M} = \sqrt{\left(\frac{\Delta Erk_M^{f}(inj)}{Erk_M^{f}(inj)}\right)^2 + \left(\frac{\Delta Erk_M^{f}(control)}{Erk_M^{f}(control)}\right)^2} \quad (23)$$

in which $\Delta Erk_M^{f}(inj)$ and $\Delta Erk_M^{f}(control)$ are given by Eq (22).

This uncertainty is indicated on the figure panels as the model error bars.

## *Otx* smFISH spot counts

Given a complete set of parameter values including $S_1$ and $S_2$, there is no uncertainty on the deterministically modeled value of *Otx* expression ($O$). However, to compare with experimental data, we computed the average of 25 values of $O$ ($O_M$), corresponding to 25 couples of ($S_1$, $S_2$) values. The uncertainty on this computed average activity ($\Delta O_M$) is the standard deviation.

Experimentally, *Otx* expression is quantified by the *Otx* smFISH spot counts. As described above (Eq (18)), we assumed that the number of fluorescent spots is a linear function of $O$:

$$Otx_{smFISH} = CO + D$$

where D is the basal value of *Otx* smFISH spot counts.

In the absence of other information, we postulated that the uncertainty on D ($\Delta D$) is the same as the standard deviation of the experimental values of *Otx* smFISH spot counts measured in the a6.8 cells. We considered that there is no uncertainty on C.

Thus, the uncertainty $\Delta Otx_{smFISH}$ is equal to $\Delta D$ for each cell, i.e. for each couple of $(S_1, S_2)$ values. In the Figs 3A, 3B, 3C, 3E, S4A, S4B, S7A and S7C the uncertainties $\Delta Otx_{smFISH}$ for the single cells are not shown, for clarity reasons.

The average $Otx$ smFISH spot counts for a given cell type is computed as:

$$Otx_{smFISH}^{M} = C \cdot O_M + D$$

with the corresponding uncertainty:

$$\Delta Otx_{smFISH}^{M} = \sqrt{\left(C \cdot \Delta O_M\right)^2 + \Delta D^2} \tag{24}$$

This uncertainty is indicated on the figure panels as the model error bars.

## Estimation of the Hill coefficients

When not mentioned explicitly, Hill coefficients were obtained by fitting the curves with Hill functions.

Since the fit of the curve $Otx$ expression (O) as a function of Erk* in the absence of the repressor (Fig 3F) was poor, we used the fact that Hill coefficients can be calculated in terms of potency as:

$$n_H = \frac{log(81)}{log(Erk^*{}_{90}/Erk^*{}_{10})} \tag{25}$$

where $Erk^*{}_{90}$ and $Erk^*{}_{10}$ are the levels of ERK activity needed to produce 90% and 10% of the maximal $Otx$ expression, respectively.

Similarly, we computed the Hill coefficient between $Otx$ expression (O) and $S_1$ in Fig 6E as:

$$n_H = \frac{log(81)}{log(S_{190}/S_{110})} \tag{26}$$

where $S_{190}$ and $S_{110}$ are the fractions of cell surface in contact with FGF needed to produce 90% and 10% of the maximal $Otx$ expression, respectively.

## Supporting information

**S1 Fig. Control of ERK activity by cell contact surfaces. A)** ERK activities in the a6.5, a6.6, a6.7 and a6.8 cell types computed with the model, shown as Erk* values (left) or as $Erk^f$ values (right), with the relation between the two given by Eq (14) with A = 1850 and B = 155.11. Results are identical to those shown in Fig 2A where only the values of $Erk^f$ are indicated. **(B)** Nuclear dpERK IF signals in individual a-line cells of NVP-treated embryos are shown as a function of the relative area of cell surface contact with A-line cells in experiments and in the model. Experimental data are shown in grey (a6.8 cell type), green (a6.6 cell type), blue (a6.7 cell type), magenta (a6.5 cell type), predictions of the model in black. To consider the presence of NVP, [ephrin] = 0.001 in the model. Hill coefficient obtained by fitting the model prediction with a Hill function: 1.96. A = 2500 and B = 256.3 in Eq (14). The area shaded in grey represents the uncertainty on the model prediction. **(C)** Nuclear dpERK signals in the a6.5, a6.6, a6.7 and a6.8 cell types in NVP-treated embryos as measured in IF experiments (left) and computed with the model (right, $Erk^f$ values). Each point represents a single cell and modeling results are computed using the measured values of $S_1$. To consider the presence of NVP, [ephrin] = 0.001 in the model. Means and standard deviations are shown in black. A = 2280 and B = 196.5 in Eq (14). **(D)** Injected/control ratios of nuclear dpERK signal in RG$\Delta$GAP injected half embryos. Left and right columns show ratios of activities of experimental dpERK

and computed $Erk^f$, respectively. Injection of p120RasΔGAP was modeled by considering $V_{rg}$ = 0.01. A = 1850 and B = 155.11 in Eq (14).
(PDF)

**S2 Fig. Experimental data and predictions for b-line cells. (A)** Left: drawing of 32-cell stage embryo, top-animal side view, bottom-vegetal side view. The ectoderm cells are encircled with a green dashed line (top) and the boundary between A- and B-line mesendoderm cells is marked with a green dotted line (bottom). The b-line ectoderm cells are colored following the color code indicated, with a-line ectoderm cells in white. The B-line mesendoderm cells are shaded with a greyscale heatmap depicting levels of FGF used in the model (A = B6.1 = 5; B6.2 = B6.4 = 0.7; B6.3 = 0.08). Middle graph shows the relative area of cell surface contact for each b-line cell with A-line mesendoderm and each B-line mesendoderm cell, with dots shaded with the grayscale FGF-heatmap. The right-hand graph shows the relative area of cell surface contact for each ectoderm cell with ephrin expressing ectoderm cells. In the graphs, each dot represents a single cell, n = 26 per cell type. **(B)** Nuclear dpERK signals in the b6.5, b6.6, b6.7 and b6.8 cell types as measured in IF experiments (left, n = 100 per cell type) and computed with the model (right, see *'Model of Erk activity in b-line cells'* section). Each point represents a single cell and modeling results are computed using the measured values of $S_1$ and $S_2$ from panel (A). Means and standard deviations are shown in black. A = 1500 and B = 144.9 in Eq (14).
(PDF)

**S3 Fig. Dual control of *Otx* expression by antagonistic transcription factors. (A)** *Otx* expression levels in the a6.5, a6.6, a6.7 and a6.8 cell types computed with the model, shown as O values (left) or as $Otx_{smFISH}$ values (right), with the relation between the two given by Eq (18) with C = 66 and D = 2.75. Results are identical to those shown in Fig 3A where only the values of $Otx_{smFISH}$ are indicated. **(B)** Relation between the concentrations of active activator of *Otx* expression ($A_p$, phosphorylated Ets1/2) and active repressor of *Otx* expression (I, unphosphorylated ERF2) and ERK activity ($Erk^*$) in the model. The Hill coefficients of the curves were computed using relation (25). When combined with Eq (15) for *Otx* expression, the relation between O and $Erk^*$ takes the form of the curve shown in Fig 3F.
(PDF)

**S4 Fig. Extended representation of the data shown in Fig 3B and 3C. (A)** *Otx* expression in dnFGFR (upper left), Eph3ΔC (upper right) and RGΔGAP (lower left) injected embryo halves. In each plot, left and right columns show experimental *Otx* smFISH spots and computed $Otx_{smFISH}$, respectively. Injection of dnFGFR, Eph3ΔC and RGΔGAP were modeled by considering $R_{tot}$ = 100, $Q_{tot}$ = 10, and $V_{rg}$ = 0.01, respectively. In Eq (18), C = 92 for the control embryos and for dnFGFR injected embryos, C = 95 for Eph3ΔC injected embryos and C = 122 for RGΔGAP injected embryos. D = 1.5 for the control and dnFGFR injected embryos, D = 2.71 for Eph3ΔC and D = 1.61 for RGΔGAP injected embryos. **(B)** Effect of the ephrin/ Eph inhibitor NVP on *Otx* expression in the four cell types. On the left, NVP-treated embryos; on the right, embryos treated with NVP and with the MEK inhibitor U0126 0.2 $\mu$M. In both cases, experimental *Otx* smFISH spots are shown on the left and computed $Otx_{smFISH}$ values on the right. Each dot represents a single cell. NVP and moderate U0126 treatment are simulated by considering [ephrin] = 0.001 and $K_{erk}$ = 0.6, respectively. In Eq (18), C = 94, D = 1.2. The data shown are the same as the ones presented in Fig 3B and 3C.
(PDF)

**S5 Fig. Asymmetric dependence of the fraction of Ras-GTP on the SOS (FGF) and RasGAP (ephrin) pathways.** Because the boundaries that can be reached by T when varying FGF or ephrin are different, ERK activity does not vary between the same limits. Thus, the relation

between Erk$^*$ and the [FGF] or [ephrin] is different in the two situations, as shown in Fig 6C. Values of the parameters used: $V_s = V_{rg} = 1$, $K_1 = K_2 = 0.5$, $K_{rg} = K_s = 1200$, [FGF] or [ephrin] = 5, $K_d = K_e = 25$, $R_T = Q_T = 2000$, $S_1 = S_2 = 0.5$, $K_b = 10^{-6}$.
(PDF)

**S6 Fig. Left: Effect of changing the $K_{MMi}$ in Eq (16) on the relation between the fraction of phosphorylated activator, $A_p$ and active ERK, Erk$^*$.** Right: Heatmaps showing the Hill coefficients of the relation between *Otx* and Erk$^*$ when changing the values of the $K_{MMi}$ in Eqs (16–17).
(PDF)

**S7 Fig. Cooperativity in *Otx* activation.** *Otx* expression levels were computed considering cooperativity in *Otx* activation (see section 'Cooperativity in *Otx* activation') Standard parameter values in Table 1 or: $K_{MM1} = K_{MM2} = K_{MM3} = K_{MM4} = 0.3$, $k_{MM3} = k_{MM1} = 14$. **(A)** Levels of *Otx* expression in the a6.5, a6.6, a6.7 and a6.8 cell types as measured by single molecule fluorescence *in situ* hybridization (smFISH, left) and computed with the model (right, *Otx$_{smFISH}$*). Each point represents a single cell and modeling results are computed using the measured values of $S_1$. Means and standard deviations are shown in black. C = 60 and D = 2.75 in Eq (18). **(B)** *Otx* expression as a function of ERK activity (Erk$^*$) in the four cell types (colored points and triangles) and computed with the model (*Otx$_{smFISH}$*, black points and triangles) considering cooperativity in *Otx* activation (see section 'Cooperativity in *Otx* activation'). In the two cases, dots indicate the control while triangles indicate the ephrin-inhibited embryos. The experimental data were fitted by a Hill function, best-fit Hill coefficient = 6.09. For the experimental points, the value of Erk$^*$ corresponding to each cell was obtained by inversion of Eq (14), with A = 3200 and B = 0. The experimental value of O corresponding to each cell was obtained by inversion of Eq (18), with C = 110 and D = 0. Modelled *Otx* outputs were computed using the experimentally measured Erk$^*$ estimates as inputs. **(C)** Levels of *Otx* expression in embryos from eggs injected with an ERF2-morpholino to inhibit translation of ERF2 (left) and modelled with I = 0.01 (Eq (15)) (right). C = 73. 4 and D = 65 in Eq (18). **(D)** Relation between the concentrations of active activator (Ap, phosphorylated Ets1/2) and active repressor (I, unphosphorylated ERF2) and ERK activity in the cooperativity model. The Hill coefficients of the curves were computed using relation (25). **(E)** *Otx* expression (O) as a function of Erk$^*$, computed with the cooperativity model considering the presence (blue line, Hill coefficient = 4.4) or the absence (orange line, Hill coefficient = 2.6) of I (ERF2). The Hill coefficients were computed using relation (25). Dashed vertical lines represent the mean values of Erk$^*$ for each cell type. **(F)** On the left: levels of *Otx* expression (*Otx*) in a6.5, a6.6, a6.7 and a6.8 cell types computed with the model when randomly choosing the values of the parameters linked to the FGF receptor pathway ([FGF], $K_d$, $K_1$, $K_s$, $V_s$, $R_T$) in the interval: standard value ± 20%. In the middle: *Otx* expression levels in the different cell types obtained when randomly choosing the values of the parameters linked to the ephrin receptor pathway ([ephrin], $K_e$, $K_2$, $K_{rg}$, $V_{rg}$, $Q_T$) in the interval: standard value ± 20%. On the right: *Otx* expression levels in the different cell types obtained when randomly choosing the values of all the parameters in the range: standard value ± 20%.
(PDF)

**S8 Fig. Heatmap showing the Hill coefficients of the relation between *Otx* and Erk$^*$ when changing the values of $k_{MMi}$ in Eqs (16–17) followed by the cooperativity model (Eq (21)).** Heatmaps showing the Hill coefficients of the relationship between, *Otx* and Erk$^*$ when changing values for $K_{MMi}$ in Eqs (16–17) using the cooperativity model (Eq (21)).
(PDF)

## Author Contributions

**Conceptualization:** Rossana Bettoni, Clare Hudson, Hitoyoshi Yasuo, Sophie de Buyl, Geneviève Dupont.

**Data curation:** Rossana Bettoni, Géraldine Williaume, Cathy Sirour.

**Formal analysis:** Rossana Bettoni.

**Funding acquisition:** Hitoyoshi Yasuo, Sophie de Buyl, Geneviève Dupont.

**Investigation:** Rossana Bettoni, Clare Hudson, Géraldine Williaume, Cathy Sirour, Hitoyoshi Yasuo, Sophie de Buyl, Geneviève Dupont.

**Methodology:** Rossana Bettoni, Géraldine Williaume.

**Software:** Rossana Bettoni.

**Writing – original draft:** Rossana Bettoni, Geneviève Dupont.

**Writing – review & editing:** Rossana Bettoni, Clare Hudson, Hitoyoshi Yasuo, Sophie de Buyl, Geneviève Dupont.

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
