## [Decision Letter · Decision Letter 0]

23 Sep 2022

Dear Dr. Dupont,

Thank you very much for submitting your manuscript "Model of neural induction in the ascidian embryo" for consideration at PLOS Computational Biology.

As with all papers reviewed by the journal, your manuscript was reviewed by members of the editorial board and by several independent reviewers. In light of the reviews (below this email), we would like to invite the resubmission of a significantly-revised version that takes into account the reviewers' comments.

We cannot make any decision about publication until we have seen the revised manuscript and your response to the reviewers' comments. Your revised manuscript is also likely to be sent to reviewers for further evaluation.

Sincerely,

David M. Umulis

Academic Editor

PLOS Computational Biology

Jason Haugh

Section Editor

PLOS Computational Biology

Reviewer's Responses to Questions

**Comments to the Authors:**

Reviewer #1: In this paper, Bettoni et al model the expression of Otx in the 32 cell stage ascidian embryo. This is where neural induction occurs in this embryo. They extensively break down a series of published experimental observations to model how the embryonic geometry, specifically the surface area contacts, between specific cells results in the activation of Erk. They then proceed to model how this activation of Erk can result in the expression (or no expression) of Otx through a minimal set of interactions between Ets and Erf.

This study is well performed. I have some minor concerns that I believe the authors should be able to address.

Minor concerns

The main criticism that the authors are surely going to get is direct comparisons with the Willaume et al 2021 Dev Cell paper. This can’t be avoided since that is the dataset they are using. Nevertheless, I would recommend removing figure 1B since it is a complete copy and paste from the previous paper and replacing it with a new schematic.

Otx is expressed in the b6.5 cells (as well as the B6.1/2/4 cells, but these are surely regulated by other processes) at the 32 cell stage. Surely the author’s model can also account for the expression in the b6.5 cells? This would do a lot to show the usefulness of the model beyond the Willaume et al 2021.

The terminology is complex and is sometimes inconsistent within the paper. If I were writing this paper, I would refer to FGF as an inducer, Ephrin as an inhibitor, Ets as an activator, and Erf as a repressor. The authors do not have to copy my terms, but can they at least look over the terms they are using, particularly with respect to Erf, where I was most confused.

Line 35 – Typo: Development biology

Line 45 – Typo: Allow us to quantify.

Line 81-83 – Ascidians are described as both “sister group” and “cousins” to vertebrates. It would probably sound better to not use two different family terms to describe the phylogeny.

Line 84 – Non-ascidian tunicates are also chordates with these properties.

Line 117 – Punctually sounds strange. It would probably sound better to remove the word.

Line 121 – Estimated ligand concentration?

Line 126 – “Shape of the ERK-Otx relation” should be rephrased.

Lines 148-152 – A lot of cell biology here that could use a reference or two.

Line 168 – In the following section?

Line 197-203 – What exactly is the basal fluorescence? Autofluorescence from the sample? Non-specific binding of the antibody? Either of these would lead to an artificially high signal. It is interesting that the model falls right in the middle of the experimental data. I think if basal fluorescence was a problem, you would see the model data at the lower end of the experimental values.

Line 224 – Would it be better to say more linear instead of “less nonlinear”?

Line 232 – Isn’t the ERK activity unchanged in the a6.8 and a6.6 cells (fig 2C)?

Line 261 – Bertrand et al 2003 should be the reference here. Weaver et al 2022 should go with the Le Gallic papers in the next line.

Fig 3D – There might not be an elegant solution, but it is hard to distinguish between the different conditions in this graph.

Line 327 – typo: intuitively logic

Line 432 – Typo: Act should be plural.

Lines 526 – I believe only FGF receptors (not Eph) were shown to be uniform through experimental overexpression. Although it makes the sentence grammatically clumsy, I think this should be stated.

Line 644 – Repression of an inhibitor is incorrect here. The inhibitor is active.

Lines 650 – Although both Ets and Erf bind to (and effectively compete for) the same DNA sites, I am not sure if considering Erf as a competitive inhibitor is appropriate. My understanding is that although concentrations are important, at any physiological concentrations, Erf will always repress until it is phosphorylated, in which state it will never repress. Does the model account for this?

Reviewer #2: In the current submission, Bettoni et al. expand on previous mathematical models to explain neural differentiation from cell geometries in the ascidian embryo. Specifically, they formulate and parametrize a full ODE model that takes cell surface areas or receptor concentrations as inputs, and predicts the resulting Otx expression state in single cells. The authors show that their model can reproduce Otx expression patterns under normal and perturbed conditions. In their model, they identify zero-order ultrasensitivity as a central mechanism to translate graded ERK activity into an all-or-none Otx expression pattern. Finally, they use the model to test the influence of different inputs such as cell surface area or the concentrations of different receptors, which leads them to make predictions for future experiments.

Overall, the paper is well written and the data are presented clearly, except some of the analyses in Figs. 5 and 6 (see below). The ODE approach used by the authors is well-established, and even though it does not yield strong unexpected or counterintuitive results, the authors use is carefully to make testable predictions. I notice that the work is closely related to the modelling aspects in a previous experimental paper by some of the same authors (Williaume et al., 2021), e.g. Eq. 16 and 17 in the present paper are the same as Eq. 22 and 23 in Williaume et al.. Some of the conclusions in the present paper, such as the stronger dependence of neural differentiation on FGF than on ephrin signaling have already been made in this previous paper, whereas others, such as the suggestion that bimodal Otx expression results from zero-order ultrasensitivity, are only developed extensively in the present manuscript. From my point of view, the merit of the current submission therefore mainly lies in meaningfully connecting different aspects of this previous work, and in characterizing the model in more - and interesting - detail.

Major points

1. How exactly has the initial parameter set been chosen? In the text, the authors only state that they used “manual fitting”. It would be informative to explain more clearly the strategy how each of the parameters has been optimized. Specifically, I am left wondering whether the parameters have initially been constrained on the wild type data only, or whether information on perturbed conditions have been used to determine parameters.

2. Related to this point, one of the main messages of the authors is that the bimodal expression of Otx is a consequence of zero-order ultrasensitivity in their model, because “the phosphorylation and dephosphorylation reactions of the two transcription factors are far from saturation” (lines 302 and 303). Is there independent evidence to set the key parameters KMM1-4 as low as they are, or have these values been chose ad hoc? In the latter case, the phenomenon of zero-order ultrasensitivity would qualify more as a prediction from their model, and should clearly be labelled as such.

3. Fig. 4: The relative changes of Erk activity and Otx expression are hard to decode from the log2-scale that the authors are using. Is it possible to indicate in this plot how big a 20% change of these quantities would be, to make it easier to identify sublinear scaling of the two activities? From the way the data is presented at the moment, I remain to be convinced that the mechanism is robust against changes in parameter values (lines 339 and 428). If I have been missing something here please clarify.

4. The heatmaps in Fig. 5 and Fig. 6 A, B are difficult to understand. Please explain in more detail what the colors mean - my interpretation is that any color value >0 indicates that the first term in the subtraction under the scalebar (nH[Erk(S1)] for Fig. 5A) has a stronger influence in increasing the Hill coefficient than the second one, but please correct me if I’m wrong here. It would be helpful if all color scale bars clearly indicated where the 0-position (no change) is.

Minor points

5. It seems to me that Fig. 1 has been entirely reproduced from Williaume et al., 2021. Even though I acknowledge the necessity of presenting this data for reference, I suggest to make it clearer that the data has been reproduced from a previous paper, and perhaps present it in a less prominent place.

6. Parameters A, B, and C, D have been changed in between panels in Figs 2 and 3. What was the reason for this adjustment?

7. Line 230: Word missing: “factor of”.

8. Fig. 3B: Can the authors stagger data from different cells to increase clarity?

9. Fig. S2B lacks a Y-axis label.

Reviewer #3: Please see the attachment for review comments

**Have the authors made all data and (if applicable) computational code underlying the findings in their manuscript fully available?**

Reviewer #1: Yes

Reviewer #2: Yes

Reviewer #3: Yes

PLOS authors have the option to publish the peer review history of their article (what does this mean?). If published, this will include your full peer review and any attached files.

Reviewer #1: No

Reviewer #2: No

Reviewer #3: No
---

## [Decision Letter · Decision Letter 1]

17 Jan 2023

Dear Dr. Dupont,

We are pleased to inform you that your manuscript 'Model of neural induction in the ascidian embryo' has been provisionally accepted for publication in PLOS Computational Biology.

Best regards,

David M. Umulis

Academic Editor

PLOS Computational Biology

Jason Haugh

Section Editor

PLOS Computational Biology

Reviewer's Responses to Questions

**Comments to the Authors:**

Reviewer #1: The authors have done an admirable job addressing my comments. I have no further concerns and I recommend the paper be published in its current form.

Reviewer #2: In the revised version of the manuscript, the authors have appropriately addressed all my points from the first round of review. They have also added new data and analysis in response to the other reviewers’ comments which all improve the manuscript. I particularly commend their approach to explore cooperativity as an alternative mechanism to obtain all-or-none OTX2 expression from graded ERK activation (new Fig. S7). Although this new data leads them to tone down their claims that zero-order-ultrasensitivity is the newly discovered reason for all-or-none OTX2 expression, I do not see this as a problem, as having these two possibilities made explicit constitutes a starting point for further experimental investigation of the problem.

The presentation of the data in Fig. 5 is now much clearer than in the original submission. When reading the text, I still had some problems following how exactly the authors draw their conclusions from the figure. What could help is to make the difference between the left and middle panels in Fig. 5A stronger (especially the change in steepness in the curve B), and explicitly state (e.g. in line 371) that a difference >>0 of the Hill coefficients as indicated by the orange color in the heatmaps indicates that the first term has stronger influence on the Hill coefficient than the second one. It took me some time to figure this out, and one could expect some readers will face the same difficulty.

Even though - as also pointed out by the other reviewers – the model in the present manuscript is still largely an extension of previous work, I believe that PLoS Computational Biology is an appropriate venue to present this in-depth analysis of modelling efforts that often does not receive the space that it deserves in more experimentally oriented journals. Therefore I strongly support publication of the manuscript in its current form.

Reviewer #3: Thanks to all the authors for their sincere efforts in addressing the comments, concerns made in the version 1. The review comment is uploaded as a separate attachment.

**Have the authors made all data and (if applicable) computational code underlying the findings in their manuscript fully available?**

Reviewer #1: Yes

Reviewer #2: Yes

Reviewer #3: None

PLOS authors have the option to publish the peer review history of their article (what does this mean?). If published, this will include your full peer review and any attached files.

Reviewer #1: No

Reviewer #2: No

Reviewer #3: No

---

## [Editor Report · Acceptance letter]

30 Jan 2023

PCOMPBIOL-D-22-00981R1 

Model of neural induction in the ascidian embryo

Dear Dr Dupont,

I am pleased to inform you that your manuscript has been formally accepted for publication in PLOS Computational Biology. Your manuscript is now with our production department and you will be notified of the publication date in due course.

With kind regards,

Zsofia Freund
